# Critical scaling of novelty in the cortex

Tiago L. Ribeiro ⓘ, Ali Vakili, Bridgette Gifford, Raiyyan Siddiqui,
Vincent Sinfuego, Sinisa Pajevic ⓘ & Dietmar Plenz ⓘ ✉

To guide behavior in uncertain environments, the brain must rapidly detect novel or unexpected events. The neocortex, involved with complex perception and decision-making, is thought to contribute to this computation, but underlying mechanisms are poorly understood. Here, we test how a few unanticipated action potentials influence local circuitry in the resting mouse visual cortex. Using targeted holographic stimulation, we evoked sparse "surprise" spikes in single pyramidal neurons and monitored their effects on hundreds of neighboring cells with 2-photon imaging. These novel spikes, distinct from the cortex's ongoing large-scale activity fluctuations, produced strong, transient recruitment, following a power-law with slope 0.2–0.3, indicating that single neurons can mobilize large fractions of the surrounding network. Ongoing activity was dominated by neuronal avalanches, highly variable, scale-invariant spike cascades characteristic of systems near criticality. Yet, the information regarding the origin of our perturbations remained reliably identifiable and distributed across most of the observed network, as shown using machine-learning classifiers. Cortical network simulations confirmed that the measured scaling and distributed information matches predictions for systems operating near criticality. These results demonstrate two hallmarks of criticality, avalanche organization and amplified responses to small perturbations, suggesting that critical dynamics enhance the cortex's ability to detect novel events.

Thriving in uncertain environments depends on the brain's ability to detect and respond to unexpected events. In mammals, such events must be rapidly and effectively communicated across brain regions to support accurate evaluation and adaptive behavior[1]. In the cortex, the fundamental unit of this communication is the action potential generated by a single pyramidal neuron. Therefore, understanding how the brain processes unexpected events requires examining how novel action potentials – those that deviate from near-future predictions based on ongoing activity – are transmitted through the surrounding cortical network.

Neuronal spiking has been found to shape cortical activity and behavior. A spike in a pyramidal neuron from superficial layers (L2/3) of the cortex of mice has been estimated to influence fewer than 1% of nearby neurons, to evoke strong local inhibition[2–5], and in reptilian cortex to recruit precise activity sequences[6]. Coordinated activity in small groups of L2/3 pyramidal neurons elicits spatially extended

variable responses[7] and can drive trained behavior via pattern completion[8,9]. Similarly, spike bursts in deep layers (L5/6) can elicit sensitized behavioral responses[10,11] and alter global brain states[12,13]. These holographic perturbation studies in mice are in line with previous findings from cortical micro stimulation in primates, highlighting the capacity of a few spikes to significantly alter brain activity under specific conditions (for review[14]). However, a comprehensive framework to quantify the impact of action potentials on the spatiotemporal dynamics of cortical networks remains elusive.

In principle, novel spikes representing unexpected events should induce propagation across the cortex both broadly (to reach multiple functional domains for evaluation, given their uncertain nature) and rapidly (to be able to act quickly). However, this dissemination of information is constrained by several factors. First, even in the resting state, the cortex exhibits substantial activity, with each neuron receiving thousands of action potentials every second. Accordingly,

Section on Critical Brain Dynamics, National Institute of Mental Health, Bethesda, MD, USA. ✉e-mail: plenzd@mail.nih.gov

individual neurons operate in a fluctuation-dominated[15], high-dimensional[16] regime that affects their ability to respond to and encode information from novel spikes[17] or sensory inputs[18]. Second, the sparse synaptic connectivity between L2/3 pyramidal neurons – estimated at only 5–8% locally and diminishing sharply within 200 µm[19–23] – further restricts the transmission of novel spikes[24]. In addition, the weak and transient synaptic currents of these failure-prone connections in vivo[2,22], coupled with strong local inhibition[25,26], further reduce the potential influence of novel spikes introduced randomly into a pyramidal neuron in the awake state. Given these limitations, understanding how novel spikes are encoded requires a focus on the network's broader dynamics rather than the activity of individual neurons or microcircuits within a particular experimental constraint. However, adopting this broader perspective presents a significant challenge due to the inherent complexity of cortical activity.

Research suggests that brain networks operating near a critical state exhibit heightened sensitivity to small perturbations[27,28], which might enable the cortex to detect and process new information despite its complex internal activity and sparse topology. In this state, the system undergoes internal fluctuations while remaining responsive to small, local events that can propagate broadly, facilitating the recognition and dissemination of novel information. In the superficial cortical layers, these fluctuations manifest as "neuronal avalanches" – scale-invariant patterns of synchronized neuron activity characteristic of the awake state[29–31]. These avalanches align with theoretical predictions of critical dynamics. However, direct experimental evidence demonstrating that novel, localized information propagates within the neuronal avalanche regime and scales in line with predictions from critical dynamics is lacking, leaving a gap in understanding whether and how criticality supports the spread of novel information in the brain.

Using an all-optical approach, we holographically triggered spikes in individual L2/3 pyramidal neurons while monitoring activity of hundreds of nearby neurons in the primary visual cortex of awake mice. Our results demonstrate robust scaling of network responses when perturbing spontaneous, ongoing avalanche activity, with novel spikes rapidly generating widespread, yet transient information-rich activity across large cortical distances. Simulations show that this pattern of response emerges naturally in networks operating near a critical state, where small inputs can lead to large effects. Together, our findings suggest that critical dynamics play a key role in enabling the brain to efficiently process unexpected information.

## Results

### Holographically triggered spikes in a pyramidal neuron induce rapid, widespread excitation in superficial cortical layers

In awake, quietly resting adult mice, we holographically targeted single L2/3 pyramidal neurons (Target Cells, TC) in primary visual cortex (V1) that chronically co-expressed the calcium indicator jGCaMP7s and the opsin ChrimsonR (~5–10 mW, 100 ms; ~100–150 trials per TC) while simultaneously monitoring spiking activity from ~100–300 nearby neurons using 2-photon imaging (Fig. 1a; 2PI, field of view: 450 µm × 450 µm; ~50–140 µm cortical depth; temporal resolution: 22 ms; $n = 6$ mice; 16 experiments). Sparse co-expression, stimulation beam profiling, motion correction analysis, and anatomical response analysis confirmed select single neuron stimulation (Supplementary Figs. 1–5).

Holographic stimulation of a single pyramidal neuron caused wide-spread, transient effects in the surrounding network (Fig. 1b–f; see also Supplementary Fig. 4). Within 132 ms (see Methods), approximately 15% of non-targeted neurons increased spiking (Positive Responders, PosR), while fewer than 1% showed decreased spiking (Negative Responders, NegR), despite high trial-by-trial variability ($\alpha = 2.5\%$ significance threshold, one-tailed Welch's $t$ test; see

Methods). Ongoing, baseline activity was consistent across all groups (TC, PosR, NegR, and non-responders, NonR), with neurons exhibiting spike count variability well above Poisson expectations (Fig. 1g, Base; $n = 54$ TC; $n = 2182$ unique neurons, classifying into $n = 1221$ PosR + 69 NegR + 7687 NonR = 8977 neurons due to multi-target experiments). Upon stimulation, PosR firing rates and spike count variance increased markedly, remaining well above expectations for a Poisson process as reported previously[32] (Fig. 1g, Resp).

### Novel spikes scale network responses without change in overall network variability

Holographically triggered spikes in TC neurons were classified as "novel" because they were not predicted by ongoing activity, as demonstrated by autocorrelation analysis. During the baseline period, ongoing spiking exhibited significant temporal order, with high auto-correlation across all groups ($r \approx 0.2$, Fig. 2a, b; Base' vs. Base). However, during the response, the autocorrelation for TC neurons (Base vs. Resp) dropped to nearly zero ($r \approx 0$; $p < 10^{-3}$, Wilcoxon rank sum), indicating that stimulation introduced spikes independently from the preceding baseline, disrupting the temporal organization of ongoing TC spiking. In contrast, temporal correlations remained stable for PosR and NonR, while the observed decrease in NegR autocorrelation reflected floor effects on suppressed low baseline activity (Fig. 2b).

Spike count changes in TC in relation to other neuron categories exhibited distinct differences for baseline and response, which we quantified using cross-correlations and scaling functions. During baseline, TC spike counts ($TC_{SC}^{base}$) were significantly correlated with all groups (average cross-correlation $<cc> \approx 0.05$; Fig. 2c) and followed a consistent scaling relationship:

$$PosR_{SC}^{base}, NegR_{SC}^{base}, NonR_{SC}^{base} \sim \left( TC_{SC}^{base} \right)^{h^{base}} \quad (1)$$

with $h^{base} \cong 0.5$ (linear regression; Fig. 2d). This scaling, which covered most spontaneous fluctuations from 1 to ~7 spikes/132 ms during baseline in single neurons (transient firing rate increases of 8 – 53 Hz) was found both for the total spike count per neuron category and per-neuron spike count (Supplementary Fig. 6a).

During stimulation, only the excitatory response in $PosR_{SC}$ showed scaling,

$$PosR_{SC}^{stim} \sim \left( TC_{SC}^{stim} \right)^{h^{stim}} \quad (2)$$

with a corresponding exponent of $h^{stim} \cong 0.3$ significantly lower compared to baseline (Fig. 2e, f; neuron: $0.26 \pm 0.09$; network: $0.28 \pm 0.10$; fit $\pm$ 95% confidence interval; see Methods; see also Supplementary Table 1). In contrast, $NonR_{SC}$ exhibited no scaling, while $NegR_{SC}$ showed negative scaling (see also Supplementary Fig. 6b). These holographic manipulations stayed within ongoing fluctuations in cortical firing as our 1 – 7 added spikes per 132 ms closely match the 1 – 7 spikes/132 ms seen during baseline (< 20% of trials exceeded 7 spikes/132 ms). We also observed qualitatively similar scaling when measuring responses directly from the calcium transients (see Supplementary Fig. 6c).

The distinct scaling relationships between TC stimulation and the excitatory network response were mirrored in corresponding positive, zero, and negative cross-correlations between TC and the other neuron categories (Fig. 2c, right). Novel spikes preferentially scale with neighboring sub-networks composed of PosR that demonstrate a significantly higher baseline correlation compared to NonR and NegR (Fig. 2c, left). On average, though, positive cross-correlation found for $TC_{SC}^{stim}$ and $PosR_{SC}^{stim}$ exhibited a relatively weak overlap in evoked and functional connectivity (see Supplementary Fig. 7; $R^2 = 0.018$, linear regression; $p < 10^{-5}$, for the $t$-statistic of the two-sided hypothesis test).

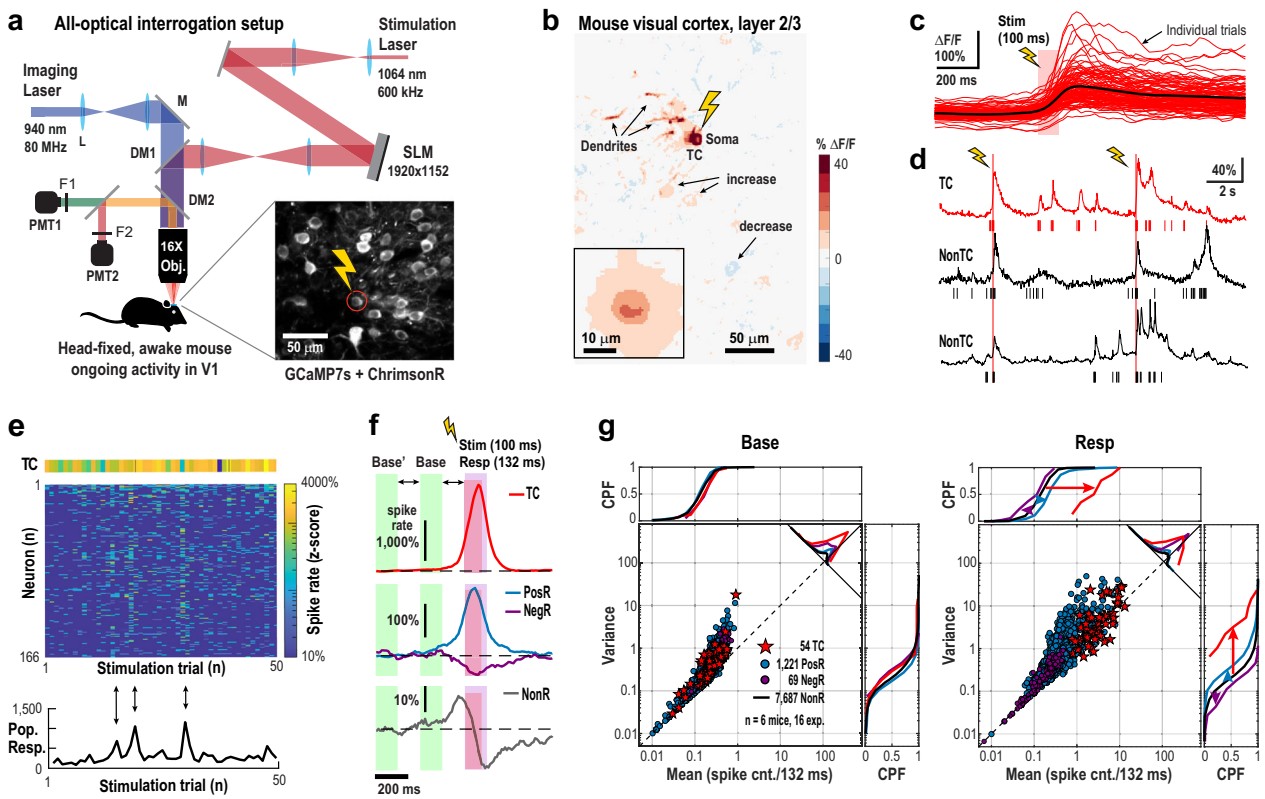

**Fig. 1 | Single-neuron holographic stimulation drives widespread excitation in superficial cortex of awake mice. a** Schematic of all-optical setup combining holographic stimulation and 2-photon imaging (2PI) in head-fixed, awake mice. Recording of ongoing activity in primary visual cortex (V1) and stimulation of single pyramidal cells were carried out during quiet wakefulness, absent visual stimulation (*DM*: dichroic mirror, *M*: mirror, *F*: filter, *PMT*: photomultiplier, *L*: lens, *SLM*: spatial light modulator). *Inset*: Mean 2P image in L2/3 co-expressing jGCaMP7s and ChrimsonR; one pyramidal neuron (*circle*) was targeted (*flash*). **b** Stimulation of the target cell (*TC*) evokes relative fluorescence changes (*ΔF/F*) in soma, dendrites, and nearby neurons (*arrows*). Averaged over *n* = 150 trials. *Inset*: Mean across 54 TCs. **c** Individual, variable ΔF/F responses (*red*) in a single TC to 100 stimulations of 100 ms duration. *Red bar*: stimulation window (*Stim, flash*). *Black*: average. **d** Example ΔF/F time course (*lines*) and corresponding deconvolved spikes (*bars*) in the TC from (**c**) (*red*) and 2 responders (nonTC; *black*). **e** Z-scored spike counts in TC and surrounding neurons (*n* = 166) across 50 trials. *Bottom*: summed trial-wise z-scores showing variable network recruitment (e.g., *arrows*). **f** Transient, mean responses to TC stimulation. Average z-scored spike rate changes in TCs (*n* = 54), Positive Responders (PosR, *n* = 1221), Negative Responders (NegR, *n* = 69) and Non-significant Responders (NonR, *n* = 7687) from 16 experiments (6 mice). *Red bar*: 100 ms stimulation window (*Stim*). *Purple bar*: 132 ms response window (*Resp*). Note that reconstructed protracted responses are due to deep interpolation (see also Supplementary Figs. 8, 10). *Base, Base'*: respective time windows for ongoing activity analyses. *Double arrows*: equal temporal spacing of respective analysis windows. **g** The response to single TC stimulation is largely excitatory, while maintaining variability above Poisson-level expectation (*dashed line*). Summary scatterplot of variance vs. mean spike count/132 ms for baseline and response (*n* = 6 mice, *n* = 16 experiments). Approximately 15% of neurons exhibit a significant increase in spike count from baseline (*PosR*), while fewer than 1% show decreased firing (*NegR*). For *NonR*, only CPF are shown. *CPF*: cumulative probability function.

---

Thus, scaling to novel spikes engages both established and alternative functional connectivity.

Scaling to novel spikes was robustly observed for different significance levels (*α*) employed. At ~10 times higher *α* (less strict significance criteria), the scaling exponent $h^{stim}$ for $PosR_{SC}^{stim}$ was still close to 0.2 (Fig. 2g). Despite our partial control of holographical spike generation due to variability from external (e.g., head movement) and internal (e.g., network inhibition) factors, these scaling relationships, nevertheless, derived a posteriori by averaging trials, confirm that driving a single pyramidal neuron distinctly affects neuronal populations compared to baseline. The scaling is also robust when using non-denoised (no Deep Interpolation) datasets (see Supplementary Fig. 8a).

The sparse synaptic connections between local L2/3 pyramidal neurons – estimated at only 5–8% and largely absent beyond 200 μm[19–23,33] – suggest that among 200 neurons observed, less than 16 neurons are synaptically connected to a given TC. Indeed, PosR were consistently found near TCs within a ~200 μm radius as reported

previously[4], in line with expectations for locally connected neighbors providing the strongest spike count increases. On the other hand, a substantial portion of PosR was distributed throughout the network (Fig. 2h), and this proportion increased as the significance threshold (*α*) was lowered (higher *α*). At the lowest significance level examined, nearly half of all recorded neurons showed significant positive responses, with an average scaling exponent of $h^{stim}$ ~ 0.22.

Recruitment of both PosR and NegR from the NonR pool (Fig. 2g–i) indicates that novel spikes are transmitted through the network via multi-synaptic pathways, triggering both increases and decreases in spiking, similar to what has been observed when stimulating groups of pyramidal neurons together[7]. When correcting for multiple comparisons[34,35], we find that our significance threshold (*α* = 2.5%) underestimates the number of significant PosR neurons by a factor of ~1.7 times, while NegR neurons become statistically non-significant (Fig. 2j). This demonstrates that novel spikes primarily drive a net excitatory response across the network. Notably, activating a single pyramidal neuron engages many neurons spread across a wide

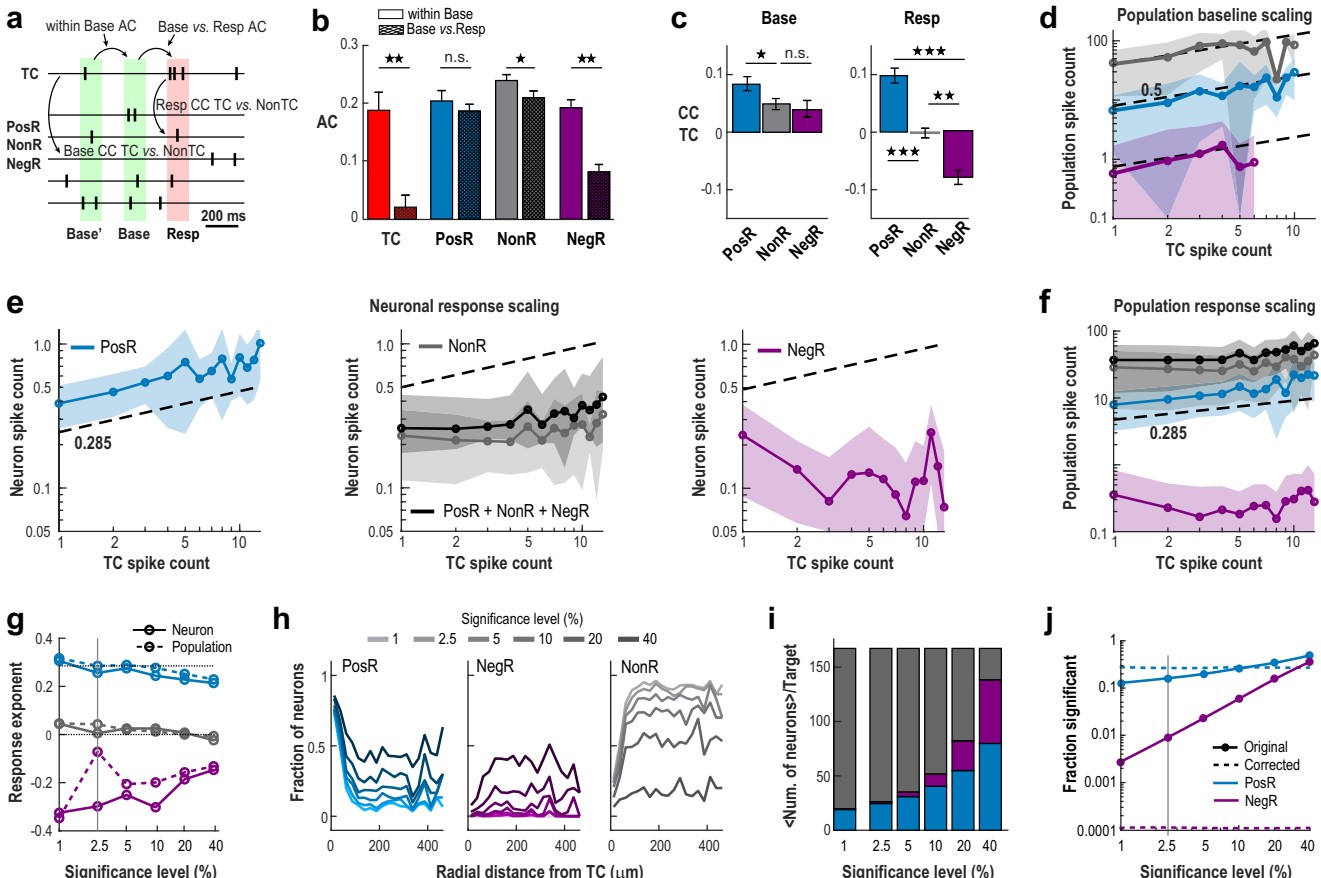

**Fig. 2 | Scaling of cortex responses to novel spikes induced in single pyramidal neurons of awake mice. a** Schematics of analysis windows used to calculate auto-(*AC*) and cross-correlations (*CC*). **b** Holographically induced TC spikes are novel, as evidenced by their zero autocorrelation with ongoing activity, while other categories maintain high autocorrelation, except for spike failure-prone NegR. **c** Base TC spike fluctuations show positive correlations with all categories. Evoked TC spikes, however, correlate positively with PosR, show no correlation with NonR, and correlate negatively with NegR. **d** Baseline scaling of spike fluctuations in TC with population spike count for each category. Note the limited range in spontaneous TC count fluctuations. Same color code as panel (**e**). **e** Novel TC spikes scale with PosR but not NonR or NegR. Summed categories added for comparison (*black*). **f** Scaling of response population spike count for each category separately and combined (*black*). **g** Positive scaling with PosR persists even when the significance level α is greatly reduced (larger α). Change in scaling exponent as a function of α per neuron (*solid lines*) and population (*dashed lines*) for each category. **h** PosR neurons are primarily located within a 100 μm radius of the TC, though they distribute throughout the network as α increases. **i** Relative number of neurons per category varies with α, with PosR and NegR constituting the majority at the lowest significance level where robust scaling is still observed. **j** Fraction of significant responders as a function of α. *Solid/dashed lines* represent original/FDR-corrected data (see Methods). *Vertical line* indicates α level used for most analysis. *Shaded areas* (**d–f**): mean ± SD across *n* = 54 experiments; *Error bars* (b, c): mean ± SE across *n* = 54 experiments. For correlation comparisons (**b**, **c**), 1, 2 & 3 stars indicate $p < 0.05$, $10^{-3}$ and $10^{-6}$, respectively, using a one-tailed Wilcoxon rank sum test (b: TC $p = 6.3 \times 10^{-5}$, NonR $p = 0.015$, NegR $p = 2.3 \times 10^{-4}$; c Base: PosR vs. NonR $p = 0.013$; c Resp: PosR vs. NonR $p = 2.6 \times 10^{-10}$, PosR vs. NegR $p = 1.1 \times 10^{-9}$, NegR vs. NonR $p = 5.8 \times 10^{-4}$).

area – an outcome that contrasts with what one would expect based on anatomical connections and synaptic physiology in activity regimes dominated by spontaneous fluctuations.

The robust scaling seen during TC stimulation may be partly due to a reduction in spiking variability, known to occur during sensory stimulation[36] and often measured using the Fano Factor (FF), which compares the variance to the mean of spike counts. We found that repeated holographic stimulation did lead to lower FF in TC neurons, consistent with more regular input. But for PosR and NonR neurons (defined at the original α = 2.5% threshold), FF did not change, even when we adjusted for differences in their average mean spike count[37] (Fig. 3a). Likewise, there was no change in FF observed for PosR and NonR population responses (Fig. 3b). To test whether variability in TC spike triggering masked changes in FF for PosR and NonR, we analyzed subsets of trials with high or low TC spike count (Fig. 3c), for overall network activity and response timing. Again, FF stayed consistently high and unchanged in both subsets under all conditions (Fig. 3d and Supplementary Fig. 9).

Our findings show that novel spikes from a single pyramidal neuron trigger strong, scalable responses in many nearby neurons, even though overall network activity remains dominated by fluctuations.

## Precise origin information from novel spikes spreads transiently across the cortical network

We next examined whether the broad network responses to novel spikes also encoded information about their origin. Across seven experiments, we stimulated up to 10 TCs in a semi-random sequence (each for 100 ms with 2–10 s between trials; Fig. 4a). We found that the origin of each novel spike event could be decoded with high accuracy from the activity of non-stimulated neurons (nonTC) using two machine learning approaches: XGBoost[38] (XGB) and Random Forest[39,40] (RF) (see Methods).

Both algorithms achieved ~60% overall decoding accuracy, and per-class *F1*-scores (harmonic mean of recall and precision) often exceeded 80% when utilizing the full population of nonTC (Fig. 4b and

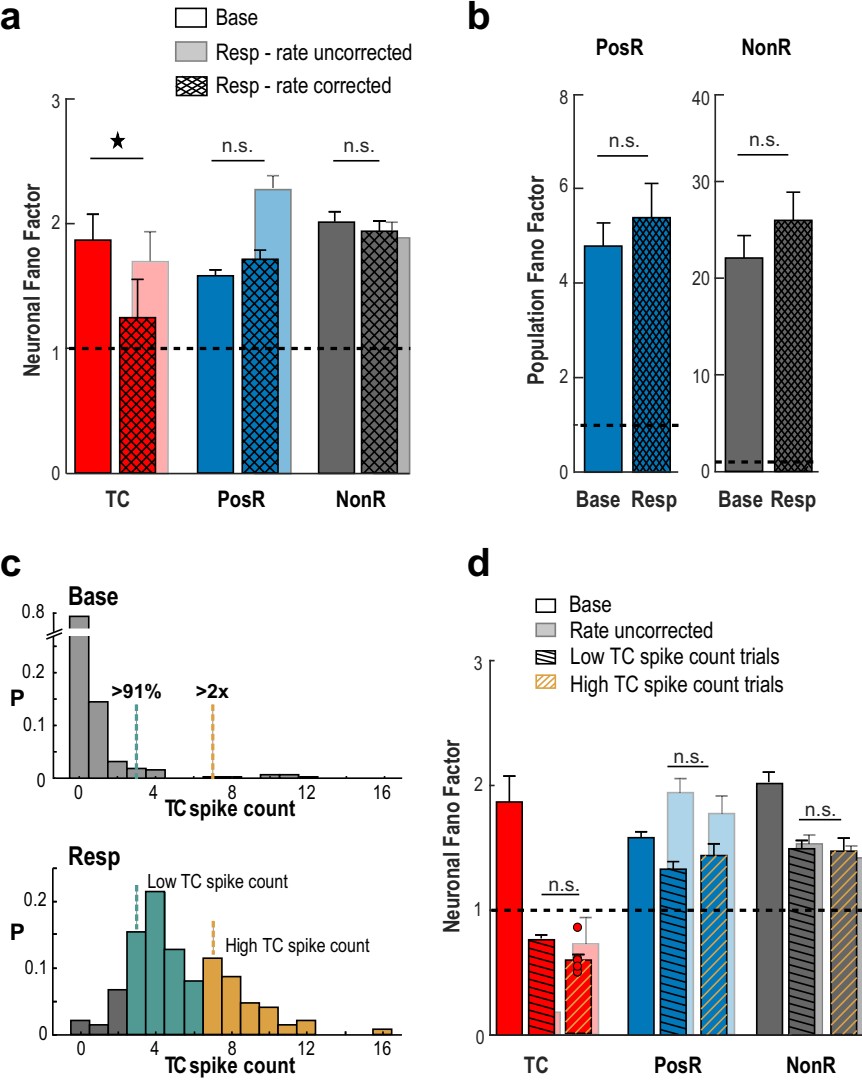

**Fig. 3 | Network variability remains unchanged despite holographically triggered spikes. a** The Fano Factor (FF) remains unchanged for PosR and NonR despite significant scaling in PosR ($\alpha = 2.5\%$). *Translucent bars*: Mean spike count (*Rate*) uncorrected FF (see Methods). **b** Population FF also shows no change across categories. **c** Example TC with trials split by low vs. high spike count. *Top*: Base spike count distribution defines 91st percentile and 2 × threshold. *Bottom*: These thresholds are used to classify Resp trials. **d** Maintained FF during Resp does not originate from variability in holographic stimulation success. Summary statistics of mean TC trials separated by low and high TC spike counts. Note the remaining high FF of PosR and NonR despite the large drop in TC FF. *Error bars* (**a, b, d**): mean ± SE across experiments, with $n = 54$, except for: (**a**) TC $n = 19$, PosR $n = 53$; (**b**) PosR $n = 53$; (**d**) TC Low $n = 18$, TC High $n = 7$ (individual points represented on the bar), PosR Low $n = 49$, PosR High $n = 38$, NonR Low $n = 53$, NonR High $n = 43$. Star indicates $p = 2.8 \times 10^{-5}$, one-tailed Wilcoxon rank sum test.

Supplementary Fig. 10a). Decoding performance positively correlated with the number of trials in which the TCs were successfully activated (Supplementary Fig. 10b). For comparison, spontaneous spikes in TC occurring within 132 ms during ongoing activity (base) could also be decoded although with significantly lower accuracy (Fig. 4c; see Methods). Remarkably, it took the removal of 10 – 70% of the most predictive neurons (ranked by Shapley analysis[41]; see Methods) for accuracy to drop to chance levels (Fig. 4d). In a representative experiment (Fig. 4e), *F1*-scores of TCs dropped to chance levels after removing > 70% of the best-encoding neurons. On average, 20– 30% of the best-encoding neurons were sufficient to achieve high decoding accuracy, regardless of the algorithm used (Fig. 4d and Supplementary Fig. 10c).

Positively responsive neurons (PosR, defined at $\alpha = 2.5\%$) played a major role, contributing 30–60% of the decoding power (Fig. 4e). To confirm that decoding wasn't driven by direct stimulation of nearby cells, we performed a spatial exclusion analysis. Excluding neurons within 10 μm of TCs had no effect on performance, but gradually removing more distant neighbors led to a steady drop in accuracy

(Fig. 4f), confirming a spatially distributed response in the network to stimulation.

Individual neurons could significantly respond to stimulation from up to five different TCs, indicating overlapping functional subnetworks capable of encoding multiple spike origins (Fig. 4e, inset). Importantly, this information was short-lived – decoding accuracy dropped rapidly after the stimulation ended, in line with the brief activity profiles observed throughout the network (Fig. 4g, cf. Figure 1f; analysis performed on a single frame – see Methods). We note that elevated decoding before the stimulus onset was attributed to temporal averaging (132 ms window) and Deep-Interpolation, which enhances signal quality but slightly blurs response timing (see Supplementary Fig. 10d). Qualitatively similar results were obtained using the calcium transient directly (Supplementary Fig. 10e).

Together, our results show that superficial layers of cortex can briefly and reliably encode when and where a novel spike originates. This supports the existence of a robust, distributed coding mechanism for processing unexpected inputs in real time across the network.

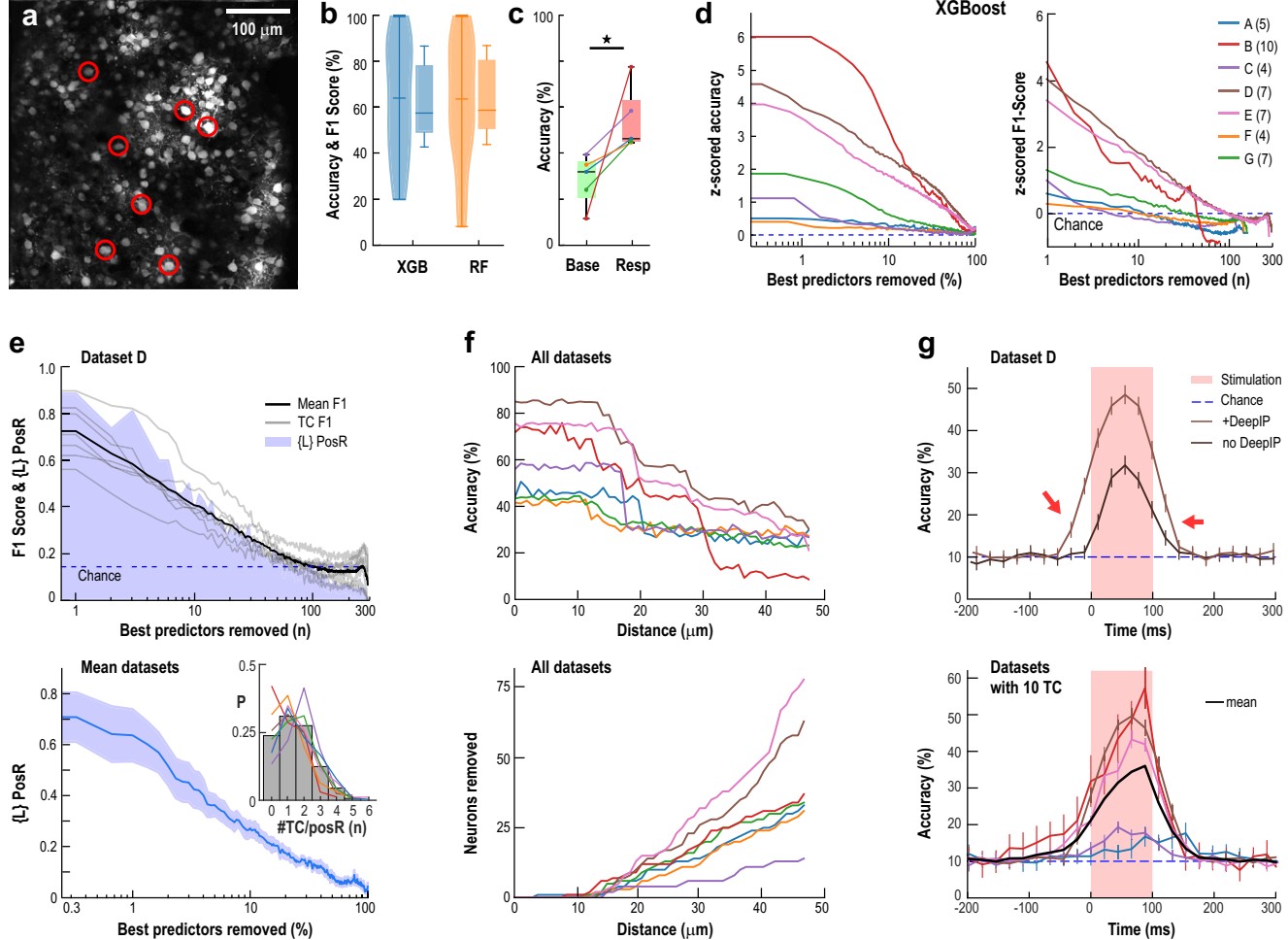

**Fig. 4 | Information about the origin of novel spikes propagates broadly, yet transiently in superficial layers of cortex. a** Example field of view showing targeted TCs (circled) and surrounding network (mean baseline activity). **b** Decoding TC identity using XGBoost (XGB) or Random Forest (RF) achieves highly above-chance accuracy (*box plots*) and per-class *F1* (*violin plots*) scores (*n* = 7 experiments; chance: 10–25%). *Box/violin plots*: *whiskers* represent min, median & max, while *box edges* represent 25th and 75th quartiles. Accuracy calculated over *n* = 140 samples, *F1* scores calculated over *n* = 880 samples. **c** Accuracy when decoding from Base compared to Resp (see Methods; *n* = 5, one-tailed paired *t* test, *p* = 0.012). **d** Distributed encoding of origin information is demonstrated by the large fraction of the top-ranked neurons (10–70% or 10–100, by Shapley value) that must be removed to reduce accuracy and *F1* to chance (dashed lines; A – G experiments with number of TC in brackets). **e** PosR contribute most to decoding. Likelihood, *{L}*, that dropped neurons are PosR. *Top*: *{L}* (*blue shade*) and *F1* scores (*gray*: single TC; *black*: mean) for a single experiment. *Bottom*: average ± SD of *{L}* for all PosR across experiments. *Inset*: Neurons can respond to multiple TCs. Distribution of the number of TC to which a PosR neuron responds. **f** Excluding neurons by radius around TCs confirms our localized stimulation paradigm limited to a single neuron. *Top*: Drop in overall accuracy when excluding all neurons within a certain distance from all TCs. *Bottom*: Number of neurons removed by this procedure. For color code, see panel (**d**). **g** Novel spike information is only transiently present in the network. Decoding accuracy rises above chance only when the response analysis window overlaps with stimulation (0–100 ms; *red shaded area*). *Top*: Single experiment ± Deep Interpolation (22-ms window). *Bottom*: All experiments with 10 TC (Chance level 10%; 22 ms window, Deep Interpolation). *Error bars*: mean ± SD over *n* = 100 replicates.

## Novel spike encoding persists across fluctuation-dominated, network state of parabolic avalanches

Our findings so far demonstrate the remarkable impact of just a few action potentials amid the thousands of spikes generated every second in the awake brain's cortical neighborhood. Given our 0.45 × 0.45 mm² field of view and ~170 imaged neurons (*cf.* Fig. 2i), we estimate our sampling to be ~5% of this neighborhood[19]. On average, we observed ~23 PosR per TC (α = 2.5%), which would translate to more than 450 neighboring neurons significantly *increasing* their firing within tens of milliseconds in response to our holographic stimulation. This estimate, however, is likely highly conservative as increasing α (lowering the significance criteria) further increases the number of neighboring neurons with detectable changes in spiking, as confirmed by our decoding results. This adjustment increasingly recruits both positive and negative responses from the pool of previously non-significant responders (it is important to note that we are likely heavily

underestimating the number of NegR[25,26], given the constraints to detect significantly lower spike counts from neurons that fire very sparsely; see Methods).

Furthermore, for many neurons, the spike count variability during novel spike encoding exceeded what would be expected from a simple Poisson process (Fig. 1), with Fano Factors comparable to those observed during ongoing spontaneous activity (Fig. 3). This high variability extended across neuronal subtypes, including positively responsive neurons (PosR). To better characterize this fluctuation-dominated regime, we analyzed its statistical properties in the context of novel spike encoding.

Holographic stimulation frequently concurred with periods of heightened ongoing activity (Fig. 5a), defined by continuous population spiking above a threshold. These active periods followed heavy-tailed distributions in both size (total spike count) and duration, consistent with previously described neuronal avalanches[29,42] (Fig. 5b).

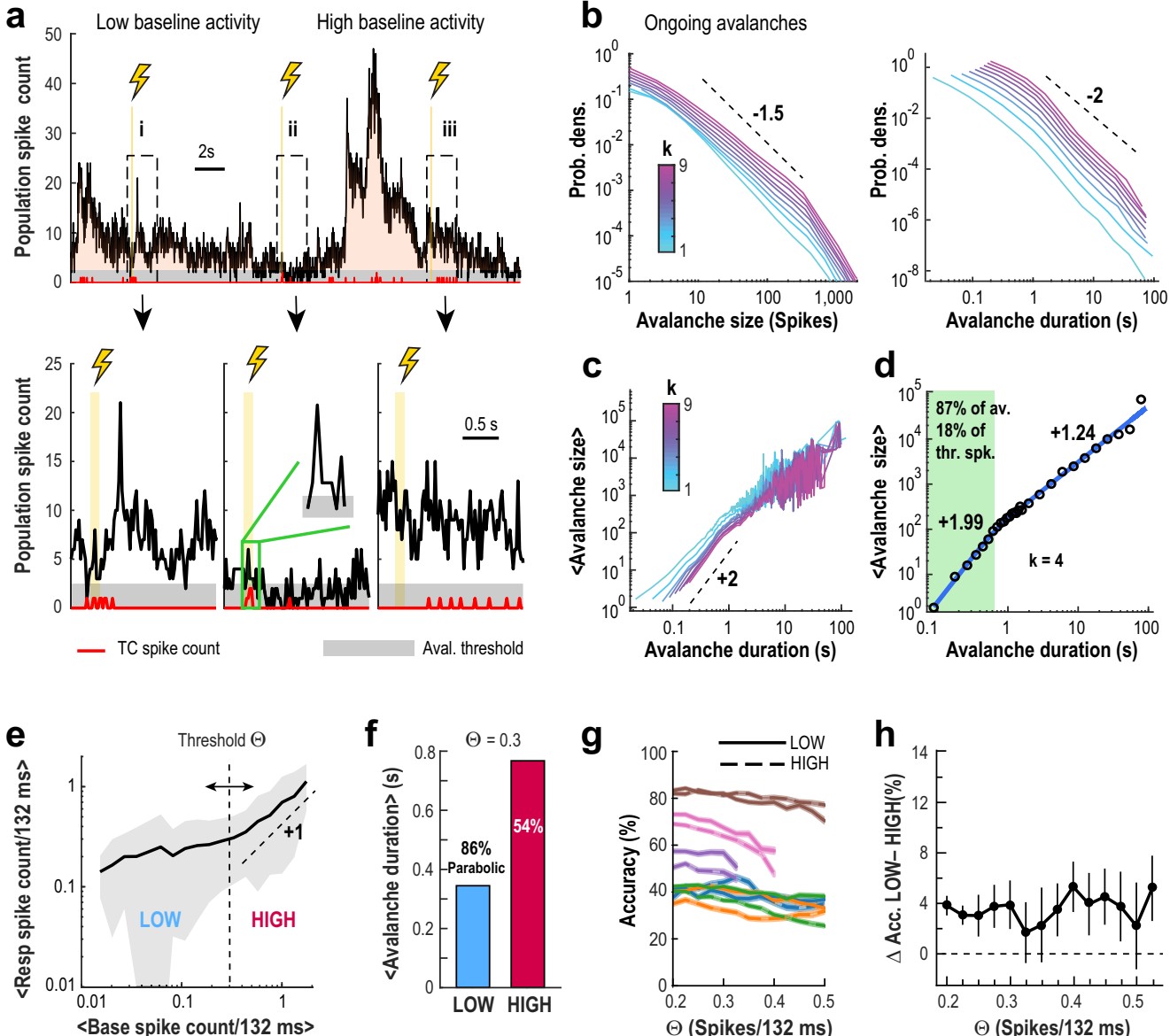

**Fig. 5 | High decoding accuracy of novel spikes coincides with parabolic neuronal avalanches exhibiting highly variable, scale-free fluctuations in baseline activity. a** *Top*: Population activity in the visual cortex of awake mice shows pronounced fluctuations during quiet resting. *Bottom*: Expanded views of population activity following three (*i – iii*) holographic stimulations of a single TC. *Vertical lines*: TC stimulation. *Red line*: TC spike count. *Gray stripe*: Population threshold for avalanche detection. Population spike count (excluding TCs) is shown per 22-ms frame. **b** Contiguous fluctuations in population activity exceeding the avalanche threshold display the characteristic features of neuronal avalanches. Shown are distributions of avalanche size (*left*) and duration (*right*) as a function of the temporal coarse-graining factor $k$ (data pooled from $n = 16$ experiments). **c** Scaling of mean avalanche size vs. duration with increasing $k$. **d** Approximately 87% of avalanches are parabolic in nature, with durations < 0.7 s and containing ~18% of thresholded spikes. At $k = 4$, scaling demonstrates a parabolic relationship (slope

~ 2) for durations up to ~ 0.7 s (*green area*), followed by a transition to more linear scaling (slope ~ 1.2) at longer durations. **e** Mean Resp spike count as a function of mean Base spike count for nonTC neurons. A threshold $\theta = 0.3$ separates LOW and HIGH baseline activity regimes; in the HIGH regime, Resp activity more closely follows Base activity (slope ~ 1). **f** Mean avalanche duration is more than twice as long for avalanches initiated during LOW baseline trials compared to HIGH. Among these, ~86% of LOW regime avalanches are parabolic (slope ~ 2), compared to ~ 54% in the HIGH regime. **g** Decoding accuracy remains consistent when trials are stratified by LOW or HIGH baseline activity, irrespective of the threshold used to define these regimes. Overall accuracy for LOW/HIGH baseline trials (*solid/dashed lines*) is shown for each dataset (color code as in Fig. 4) as a function of threshold $\theta$. **h** Difference in accuracy between LOW and HIGH baseline trials is minimal and largely independent of threshold, with LOW trials exhibiting slightly higher accuracy across $n = 6$ experiments. *Shaded areas/error bars* (**e**, **g**, **h**): mean ± SD.

After correcting for subsampling using temporal coarse graining, we found that the average avalanche size scaled quadratically with duration − a key feature of parabolic avalanches[29,42,43]. These parabolic events included avalanches lasting up to ~ 0.7 seconds, which accounted for 87% of all avalanches and about 20% of spikes during spontaneous activity (Fig. 5c, d). Longer avalanches (> 1 s) showed near-linear scaling, suggesting activity too widespread to be fully captured within the imaging field of view[29,42,44]. In contrast, this near-

linear scaling is observed for all avalanches in shuffled data (Supplementary Fig. 11).

Despite these dynamic and large-scale fluctuations, decoding the origin of novel spikes remained highly accurate across avalanche activity regimes. When baseline activity was low ($\theta < 0.3$ spikes/132 ms per neuron), stimulus-evoked spike counts scaled modestly; when baseline activity was higher ($\theta > 0.3$), the scaling approached a value of 1, indicating strong coupling between pre-stimulus and evoked activity

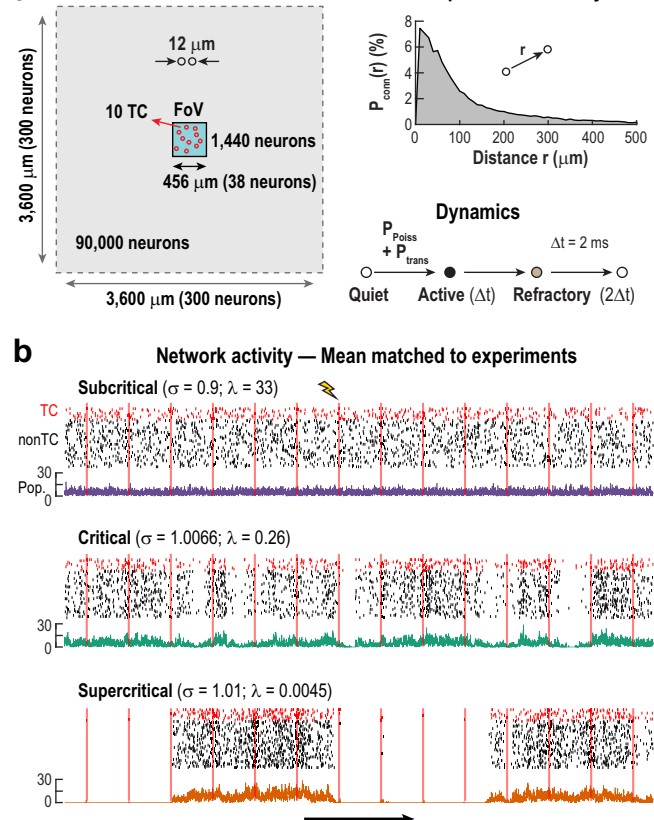

**Fig. 6 | Simulations of 2D networks reproduce experimental findings under external drive: Simulation setup. a** A 2D network of 90,000 neurons with local spatial connectivity (mean degree $\bar{k} \approx 43$ connections/neuron) receives external Poisson input. Analysis is focused on a central field of view (FoV) comprising 1440 neurons (456 × 456 μm²) and 10 target cells (TC), mirroring experimental conditions and reducing edge effects. Neurons transition through quiet, active, and refractory states, driven by fixed-rate external Poisson input and activation from neighboring neurons with fixed probability. **b** Network activity is shown for subcritical, critical, and slightly supercritical regimes. TC (*red*) and nonTC (*black*; 50 neurons randomly selected) spike trains are displayed during repeated TC stimulation (*red bars*). *Bottom*: Color-coded representation of summed population activity across the FoV illustrates overall network dynamics.

(Fig. 5e). This allowed us to classify activity into low- and high-baseline regimes, both of which prominently featured parabolic avalanche dynamics (Fig. 5f). Importantly, decoding accuracy differed by less than 4% on average between these regimes, regardless of the threshold used (Fig. 5g, h), and response scaling remained consistent across both states (Supplementary Fig. 12a).

These results suggest that cortical networks in vivo operate within a fluctuation-dominated regime where variable responses to local perturbations scale in line with expectations from critical dynamics[28,45], which we further explore in the following section.

**Model simulations of critical networks agree with the observed experimental perturbation responses**

To study how networks with varying internal excitability respond to local stimulation, we simulated a two-dimensional network of 90,000 excitatory units with local, radial connectivity $P_{conn}(r)$ and three-state dynamics (rest, active, refractory) (Fig. 6a). Analysis focused on a central field of view of 456×456 μm² with 1440 neurons, including 10 target cells (TCs), to mimic experimental access and reduce edge effects.

Each resting unit could be activated either by an external Poisson input at constant rate $P_{Poiss}$, or by an active neighbor with transmission probability $P_{trans}$. The system exhibits dynamics of the directed percolation universality class, with a phase transition centered around a critical state − between a subcritical phase (where activity fades without input) and a supercritical phase (where activity can self-sustain). We define

$$P_{trans} = \sigma / \bar{k} \qquad (3)$$

where $\sigma$ is the branching parameter − the average number of near future downstream spikes per originating spike and $\bar{k} \approx 43$ is the average number of connections per unit. Near criticality, $\sigma$ is close to 1 and adjusting $\sigma$ moves the system in and out of the critical state. Empirically, the critical point was found to be near $\sigma = 1.0066$, slightly above 1 to compensate for propagation loss (e.g., spike collision) in the network.

Figure 6b shows example traces of ongoing network activity under subcritical, critical, and slightly supercritical conditions, highlighting both the increase in overall activity and the emergence of larger fluctuations as the network approaches criticality. This shift is quantified in Fig. 7a, where the relationship between network activity and external drive ($P_{Poiss}$) reveals the critical transition. In the subcritical regime ($\sigma \ll 1$), inputs spread weakly, and network activity increases linearly with external drive, yielding a slope $\gamma \approx 1$ (see refs. [46,47]). In the supercritical regime ($\sigma > 1.0066$), even low input leads to network saturation, resulting in $\gamma \approx 0$. At the critical point, the system produces scale-invariant responses – external spikes can trigger cascades of all sizes – producing a power-law relationship with an intermediate slope $0 < \gamma < 1$. For our 2D network, the predicted critical exponent is $\gamma = 0.285$ (see ref. [28]), as shown in Fig. 7a (green curve). To match experimental conditions, we adjusted $P_{Poiss}$ in each simulation to reproduce the average baseline activity observed in our in vivo experiments.

As shown in Fig. 7b, the distribution of ongoing firing rates is narrow in subcritical conditions but broadens near the critical point, reflecting a fluctuation-dominated regime. When measuring the response on PosR as a function of TC spike count, we observed power-law scaling, with the slope approaching the one obtained experimentally ($h^{stim} \sim 0.285$) at criticality (Fig. 7c, d). Power law scaling can also be observed during baseline (Supplementary Fig. 13), although with slightly different exponents, which depend on how correlated the ongoing activity is (cp. Fig. 2d). Interestingly, when considering only trials in which the network had a low amount of activity preceding the stimulation, the slope of the scaling responses crosses the value 0.285 close to criticality, whereas considering only trials with a high amount of activity we observe that the highest slope happens at criticality, albeit at a smaller value, as in the experiments (Supplementary Fig. 12b, c). Importantly, classification accuracy improved sharply from subcritical to critical regimes and decayed gradually as increasing numbers of neurons were removed (Fig. 7e), again matching experimental observations. On the supercritical side, decoding is made trivial on many trials when the network was completely quiet preceding stimulation (see Methods). To more properly compare decoding accuracy, we therefore removed those trials from the pool (see also Supplementary Fig. 14b, c). When decoding during baseline (see Methods), accuracy remained at chance levels regardless of branching parameter (Supplementary Fig. 14a). Furthermore, decoding success was higher for targets that were more well-connected to the network (output strength estimates the expected number of spikes each TC spike would generate in a quiet network; see Methods) and was maximized at criticality (Fig. 7f).

Despite higher activity fluctuations, this critical regime maximized the number of positive responders (PosR) to TC stimulation (Fig. 8a). As in the experimental results, the fraction of cells that are

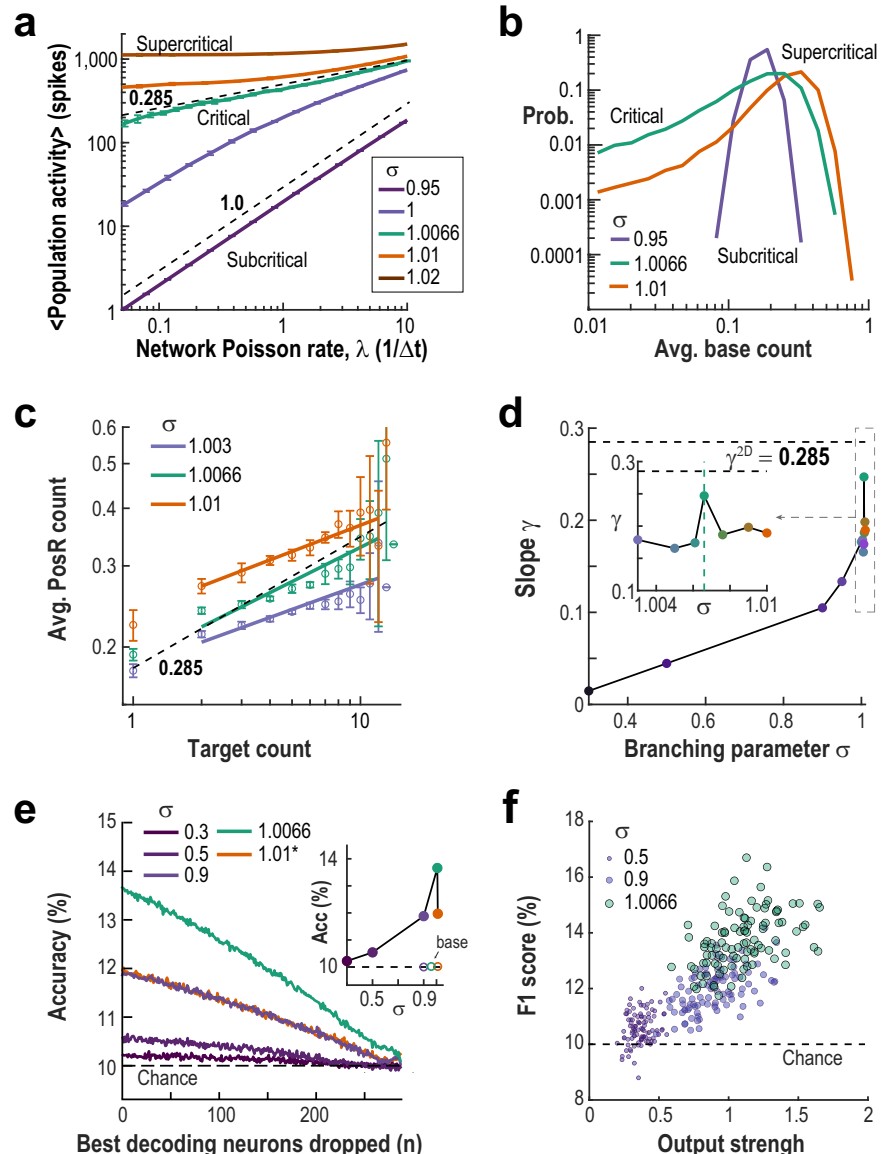

**Fig. 7 | Simulations of 2D networks near criticality reproduce experimental response scaling and decoding. a** Response scaling as a function of spike transmission probability ($P_{trans} = \sigma/\bar{k}$ (eq. 3), where $\sigma$ is the branching parameter and $\bar{k}$ is the average connectivity) reveals a power law (exponent -0.285) at the critical point (*green*). Subcritical (*purple*) and supercritical (*orange*) regimes show linear and self-sustaining behavior, respectively. *Dashed lines* indicate reference power laws with exponents 0.285 and 1. **b** Distribution of mean baseline spike counts for different regimes. Subcritical networks (*purple*) exhibit reduced fluctuation range in base*line* activity. **c** Mean ( ± SD, $n = 100$ simulated experiments) PosR response as a function of TC spike count. Near criticality, response curves follow a power law similar to experimental observations (see Fig. 2e). The *Dashed line* is a power law with exponent 0.285. **d** Power law exponents for PosR response curves across varying branching parameters. Exponents approach the experimentally observed value of

~ 0.28 close to criticality. **e** TC decoding accuracy peaks near criticality and decreases to chance only after removing a very large fraction of the top predictive neurons. Accuracy is shown as a function of the number of best decoding neurons removed for varying branching parameters. For the supercritical network ($\sigma = 1.01$), trials without spikes in the preceding baseline were removed (see Methods). *Inset*: Overall decoding accuracy during Resp (*full symbols*) shows a peak at criticality, while accuracy during Base (*open symbols*; see Methods) remains at chance levels. **f** *F1*-scores increase with TC output strength (*circles*). Output strength is defined as the sum of all probabilities that a single TC spike, in a quiet network, propagates to direct and indirect connected neighbors up to order 5. This measure estimates the expected number of spikes generated per TC spike, disregarding refractory period and spike collisions (see Methods).

PosR is maximal around TCs and decays over larger distances (Fig. 8b). At criticality, about 93% of neurons receiving direct input from TCs were classified as PosR. However, only about 16% of all PosR had a direct connection from TCs, showing the importance of multi-synaptic pathways in the critical network. This reveals a unique "mixing" effect at criticality: while a TC's average output strength is higher, its functional spread − the range of PosRs it engages − becomes much broader, reflecting the greater diversity seen in critical states (Fig. 8c).

To further examine how the network's dynamical regime affects information spread, we measured the number of PosR for each TC in a

subcritical network ($\sigma = 0.5$) and compared it to that of the critical network ($\sigma = 1.0066$). Even though the two networks have the same structure, the number of PosR for each TC is uncorrelated between them (Supplementary Fig. 15). This means that measuring a TC's effect under subcritical conditions (lower excitability) gives little or no indication of its impact near criticality.

Together, these results strongly support the idea that novel inputs are most effectively propagated in networks operating near criticality, where activity is governed by fluctuations and scale-invariant dynamics.

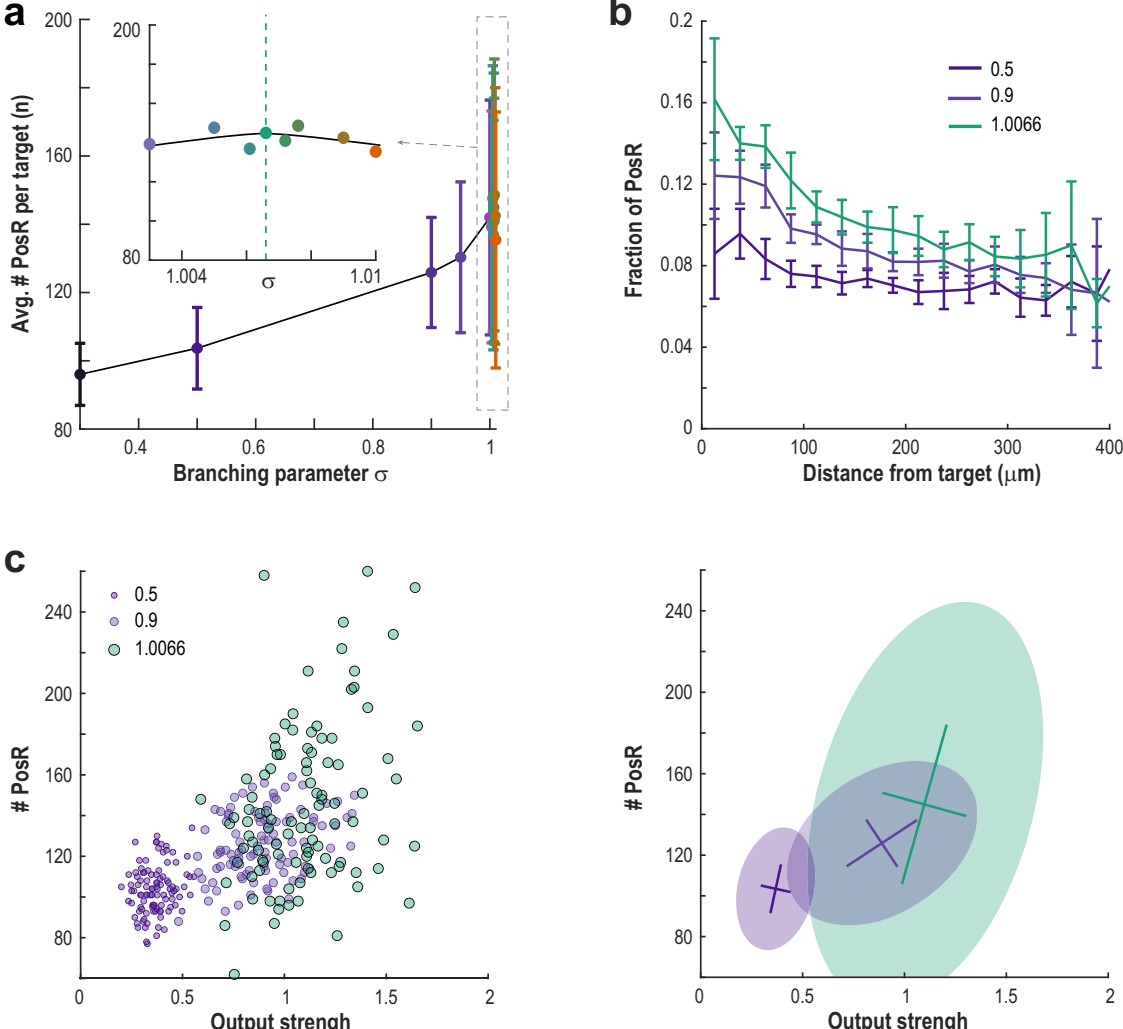

**Fig. 8 | Total number, spatial clustering and variability of PosR neurons, as well as TC output strength, peak near network criticality. a** The mean number of PosR per TC is highest close to the critical point of the network. **b** PosR neurons are most concentrated near TCs, with this spatial clustering diminishing as the network moves away from criticality. Average fraction of neurons classified as PosR as a function of distance from the TC. **c** Both mean and variability of TC output strength and the number of PosR neurons increase as the network approaches criticality. *Left*: Number of PosR per TC (*circles*) plotted against TC output strength for sub-critical and critical regimes. *Right*: Covariance matrix ellipses (*shaded area*; 2.5 SD) and center of mass (*lines*; mean ± SD) indicate shifts in mean and variance for the scatter plots on the left. Calculated over $n = 100$ unique TCs. *Error bars* (**a**, **b**): mean ± SD, $n = 100$ simulated experiments.

## Discussion

Efficient communication of novel neural signals is vital for processing surprises and guiding future actions. Using an all-optical approach allowed us to holographically induce action potentials in single neurons and simultaneously measure their effect in the network at cellular resolution and high fractional sampling of the neuronal population in the awake, quietly resting mouse. Alternative perturbation approaches, such as extracellular microstimulation, lack spatial selectivity[48], or as in the case of intracellular patching[2], are limited in capturing the network response. Here, we demonstrated in awake animals that introducing spikes in a single L2/3 pyramidal neuron rapidly activates widespread, net excitatory responses in the surrounding cortical network, encoding the novel stimulus' origin. Remarkably, these responses scaled robustly and were efficiently decoded amidst ongoing, large fluctuations in network activity organized as scale-invariant, parabolic neuronal avalanches. Our scaling exponent of ~ 0.2 – 0.3 in response to novel spikes was captured in a 2-dimensional network with critical dynamics, allowing for highest decoding accuracy and engaging large neighborhoods in encoding stimulus origin. This high susceptibility and scaling to local perturbations[27,28], suggests critical dynamics as a

cornerstone of cortical processing, offering key insights into brain function.

Our findings challenge a longstanding discussion on cortical coding fueled by the intricate statistical anatomy of the cortex and its synaptic connections. Pyramidal neurons in associative layers establish few synaptic connections with their neighbors[19–23]. These excitatory connections are relatively weak, prone to failure to transmit action potentials to a postsynaptic cell, and decay within milliseconds[22,49]. The combination of sparse and weak connectivity makes it unlikely that a spike in a single neuron reliably triggers activity propagation in the surrounding network. To address this, most cortical codes rely on activity patterns that amplify synaptic impact on postsynaptic pyramidal neurons.

For instance, coincident activation of numerous synapses from diverse neurons converging onto a postsynaptic target enhances the likelihood of activity propagation, favoring coding models based on local synchrony, such as synfire chains[50,51], waves[52,53] or phase-locked oscillations[54,55]. Alternatively, repeated or burst firing of individual neurons[12,56], strong synaptic connections[57], subthreshold priming by background activity[15,58] supported by subcellular amplification

mechanisms such as short-term synaptic facilitation[59,60] and non-linear dendritic processing[61] can dynamically increase individual connection strength supporting rate-based coding.

Our findings do not support a dichotomic separation between synchrony and firing rate to secure propagation of neuronal activity (see also refs. [24,62]). We show robust scaling of novel spikes within the cortical neighborhood from single to multiple action potentials delivered within 100 ms window without an obvious threshold above which propagation is facilitated. This suggests that neither tight synchrony nor repeat firing is strictly necessary for effective encoding of novel spikes[63]. This conclusion is in line with findings that spike counts over 100 ms predict behavior independent of considerations of synchrony at the millisecond level[64].

By emphasizing the sufficiency of a few action potentials in cortical codes, our results thus challenge existing paradigms and highlight a minimalist yet robust mechanism for cortical information processing, supporting the prominence of sparse, single action potential firing found in vivo for associative layers[65]. Furthermore, the recruitment of significant responders is most likely composed of a mixture of direct (excitation, inhibition) as well as indirect (disfacilitation, disinhibition) local circuit interactions[4,7], while preserving the general scaling relationship (cf. Fig. 2). Simulations have found that in such a fluctuation-dominated but relatively balanced regime, a reduction in membrane conductance due to increased synaptic bombardment still enables neurons to generate a few spikes in response to rapid changes in synaptic input[17].

The importance of critical dynamics for sensory cortical networks has been argued from the point of view of optimal decoding of stimulus intensities through maximization of the dynamic range[47]. In that context, a critical network is expected to produce response-input curves that scale as a power law, with exponents that are typically much smaller than one[28]. These low exponents allow the network to present distinct responses for a wide range of stimuli. On the other hand, the difference between these distinct responses gets smaller as exponents approach zero, indicating the possibility of an optimal value depending on the necessity of having a larger dynamic range versus the need for a more precise differentiation between responses. The study of psychophysics has demonstrated that psychological value scales with stimulus intensity as a power law with exponents < 1, a result known as Stevens' law of psychophysics[66]. Therefore, it has been argued that the origins of Steven's low exponents for response curves might originate precisely at this critical networks' ability to amplify low-intensity stimuli and thus respond to a wide range of stimulus intensities[67]. We were able to show that in vivo sensory cortical networks do display low-exponent, power-law behavior for their response to varying stimuli intensity (as proxied by the number of spikes evoked in a target cell during stimulation; see Supplementary Tables 1 & 2). This is in line with predictions from critical dynamics, with the observed exponents closely matching the expected value (0.285) for a two-dimensional directed percolation system[28]. The expected exponent for the response scaling of a 3D network is 0.45, whereas higher-dimensional systems would show an exponent of 0.5. Note that our observed ~ 0.5 slope during baseline activity does not represent a response of the network: since neuronal firing during ongoing activity is correlated, that relationship is more indicative of autocorrelation in the network as a whole. Our measured scaling exponent thus emphasizes the intriguing finding that superficial layers 2/3, defined anatomically as an extended 2-dimensional sheet, functionally also operate more closely to a 2-dimensional critical network.

Our findings of maintained high Fano Factor in evoked responses is in line with the expectation for a critical system, though, in contrast to the quenched variability demonstrated in response to sensory stimuli[36]. This difference may arise because single-neuron stimulation introduces a smaller perturbation than sensory stimulation, which in the awake state is dominated by inhibition[25], affecting response variability. Early studies in anesthetized animals[18] (e.g., in the visual cortex of cats) demonstrated that trial-by-trial variability in stimulus responses can be approximated as average response overlaid on a correlated, variable background. In the awake state, our observation of consistently high FF values for responses in the presence of scale-invariant parabolic avalanches indicates the need for a conceptual approach that bypasses the reliance on averaging. Avalanches emerge in superficial layers of cortex when animals recover from anesthesia[68,69], exhibit high neuronal selectivity[29,70] and spatio-temporal order[71–75] characterized by power law statistics with exponents ≤ 2 that is, averages and variability are not defined. Yet, we demonstrate that decoding of novel spikes is similar across the fluctuation-dominated avalanche regime. We note that, because our avalanche analyses were computed within finite imaging epochs and fields of view, the duration and size distributions are truncated and can be biased by these finite-window effects. Therefore, their corresponding estimated exponents should be interpreted with caution.

Our decoding analysis provided important insights into the fine-grained nature of this communication of novelty. First, strongly decoding neurons were found in the immediate neighborhood of TCs, supporting a spatial weighing in response, in line with expectations from the local, synaptic connectivity statistics of pyramidal neurons. On the other hand, a large fraction of neurons with high decoding contribution did not fall into the PosR category or were located at long distances from the TC, supporting the contribution of indirect pathways. Similarly, in our simulations, the number of identified PosR greatly surpassed that of direct postsynaptic neurons of each TC, further suggesting the importance of indirect pathways to the coding of novel information. In fact, only about 16% of PosR had a direct connection from TCs. This number could be even higher for the brain, since the presence of inhibitory connections (absent in our simulations) could facilitate certain pathways in detriment of others, while also requiring stronger excitatory-to-excitatory connections in order to stay in a balanced propagation regime. This also de-emphasizes reliance on distinct, non-overlapping network connectivity. We note that our model is very simple and does not capture the depth of the dynamics observed in the brain. Therefore, it was expected it would not reproduce all results observed. For instance, our model requires fine-tuning to the critical point and presents highly sensitive scaling and decoding accuracy with a slight change in dynamical regime. Our experimental results indicate a more robust behavior, with exponents varying little as a function of activity level, for example, as well as much higher decoding accuracy. Adding more complexity to the model, such as the presence of inhibitory neurons, could help bridge this gap.

The ability of PosR neurons to contribute significantly to the decoding of multiple TCs is in line with the spatial overlap of neuronal populations encoding different stimuli and their temporal separation. Our findings align with the notion that cortical networks are organized to maximize coding efficiency and redundancy, where overlapping populations can fluctuate in their response patterns to encode different stimuli[76,77]. The rapid decay found for the encoding of novel spikes further underscores this dynamic, ensuring that novel signals do not disrupt ongoing network activity but still convey novel information effectively within the constraints of the temporal and spatial coding architecture described for e.g., visual stimuli of the primary visual cortex in the awake mouse[78].

Our results reveal the communication of single spikes within the scale-invariant, critical-state dynamics of cortical networks, where synchronized fluctuations are neither overly constrained nor entirely random. We propose this remarkable sensitivity of the mammalian cortex to efficiently disseminate novel information from individual neurons underscores the fundamental role of critical dynamics in cortical communication and computation.

## Methods

### Animal surgery

All procedures were approved by the NIH Animal Care and Use Committee (ACUC) and experiments followed the NIH *Guide for the Care and Use of Laboratory Animals*. Mice were obtained from Jackson labs, bred inhouse with C57BL/6 backgrounds (Jackson Laboratory) under a reversed 12:12 h light/dark cycle. Chronic 2PI in adult ( > 6 weeks) mice was enabled by using a head bar in combination with a cranial window placed centered at ~2.5 mm from the midline (right hemisphere) and ~1 mm rostral to the lambdoid suture and consisting of a stack of circular glass cover slips using established protocols[79]. Mice were injected with a 7:1 mixture of a viral construct to express jGCaMP7s[80] and ChrimsonR[81] in pyramidal neurons using the CaMKII promoter. A total of 2 – 3 injections of virus (100–400 nL; < 1 μL in total; $10^{13}$ vg/mL; pGP-AAV-syn-jGCaMP7s-WPRE AAV9 & pAAV-CamKIIa-ChrimsonR-mScarlet-KV2.1, Addgene) were administered into the right hemisphere. Chronic 2PI started after > 2 weeks in identified V1 at a depth of ~100–200 μm. To prepare for retinotopic mapping, and 2PI imaging and stimulation, mice were briefly anesthetized (isoflurane ~2%, < 2 min) and then quickly moved over to the recording platform. After fixing the animal's head using a head bar mount, the mouse was allowed to recover from anesthesia for ~30 min before recordings started.

### Identification of V1 maps

Retinotopic maps of V1 and higher visual areas (HVAs) were generated for all mice prior to recording using published protocols[82,83]. Briefly, awake, head-fixed mice faced with their left eye a 19" LCD monitor placed at 10 cm distance and tilted 30° towards the mouse's midline. Using Psychophysics toolbox[84], contrast-reversing, spherically corrected checkerboard bars were drifted across the screen vertically (altitude) and horizontally (azimuth) for each of the four directions (30 repeats per direction). Simultaneous wide-field imaging (Quantalux, Thorlabs) captured jGCaMP7s fluorescence, which was averaged for each direction. Altitude and azimuth phase maps were calculated by phase-wrapping the first harmonics of the 1D Fourier transform for each of the four averages and subsequently subtracting the maps of the opposite directions[83]. Sign maps were generated by taking the sine of the angle between the gradients in the altitude and azimuth maps and processed[82]. Borders were drawn around visual area patches and overlaid onto anatomical reference images to identify V1.

### Holographic stimulation of single neurons

Holographic stimulation[85] was done by manually selecting individual pyramidal neurons as target from the 2-photon image of the field of view in both red and green channels. Neurons that express the opsin (ChrimsonR, with mScarlet tag) were selected for stimulation. Based on the selected targets, a map of top-hat patterns with diameter of 10 μm was used to generate the hologram on the Spatial Light Modulator (SLM, Meadowlark, LCoS high-resolution 1920 × 1152 phase modulator). Light from the laser/OPA system (Carbide/Orpheus-F, Light Conversion; $\lambda = 1064$ nm, repetition rate = 600 kHz, pulse width < 100 fs) was expanded (4x) to fill the area of the SLM (8 mm × 8 mm). The hologram on the pupil plane was shrunk to fit the size of the Galvo-Galvo scanners (~3 mm diameter) and form the desired pattern of top hats after the objective lens (Nikon 16X, working distance = 3 mm and effective focal length = 12.5 mm) on the selected targets. Each target neuron was continuously stimulated by the top-hat beam shape for 100 ms for a total of ~100 − 150 trials per experiment, with an interval of 2 s (10 s in some recordings) between trials. Power at target neurons ranged from ~2.5 − 10 mW. It is important to highlight that no artifact from this stimulation was detectable on our imaging with our combination of GECI/opsin, stimulation wavelength and the low power required to stimulate single targets (see Supplementary Fig. 4).

### 2PI, pre-processing pipeline and meta data collection

For standard 2PI, images were acquired by a scanning microscope (Bergamo II series, B248, Thorlabs Inc.) coupled to a pulsed femtosecond Ti:Sapphire 2-photon laser with dispersion compensation (Chameleon Discovery NX, Coherent Inc.). The microscope was controlled by ThorImageLS and ThorSync software (Thorlabs Inc.). The wavelength was tuned to 940 nm to excite jGCaMP7s. Signals were collected through a 16 × 0.8 NA microscope objective (Nikon). Emitted photons were collected through a 525/50 nm band filter using GaAsP photomultiplier tubes. The field of view was ~450 × 450 μm². Imaging frames of 512 × 512 pixels were acquired at 45.527 Hz by bidirectional scanning of a 12 kHz Galvo-resonant scanner. Beam turnarounds at the edges of the image were blanked with a Pockels cell. The average power for imaging was < 70 mW, measured at the sample.

The obtained tif-movies in uint16 format were rigid motion-corrected via the python-based software package '*suite2p*'[86]. Registered images were further denoised using machine-learning based, deep interpolation[87] (see below) and then semi-automatically processed by suite2p for regions of interest (ROI) selection and fluorescence signal extraction. We performed visual curation of the automatically selected ROIs to ensure only neurons were included in the analysis; this is done so processes, such as the dendrites identified in Fig. 1b, are not part of the analysis and therefore cannot be classified as significant responders or contribute spikes to response curves or other measurements. For each labeled neuron, raw soma and neuropil fluorescence signals over time were extracted. Spiking probabilities were obtained from neuropil-corrected fluorescence traces ($F_{corrected} = F_{ROI} − 0.7*F_{neuropil}$) via MLspike (https://github.com/MLspike), utilizing its autocalibration feature to obtain unitary spike event amplitude, decay time, and channel noise for individual neurons.

### Deep-interpolation

Deep-interpolation[87] (Deep-IP; https://github.com/AllenInstitute/deepinterpolation) removes independent noise by using local spatiotemporal data across a noisy image stack of $N_{pre} + N_{post}$ frames to predict, or interpolate, pixel intensity values throughout a single withheld central frame. The deep neural network is a nonlinear interpolation model based on a UNet inspired encoder-decoder architecture with 2D convolutional layers where training and validation are performed on noisy images without the need for ground truth data. As described previously in detail[29], after rigid motion correction, individual denoised frames were obtained by streaming one 60-frame ($N_{pre} = N_{post} = 30$ frames) registered, image stack through the provided Ai-93 pretrained model[87] for each frame to be interpolated. At an imaging rate of ~45 Hz, these 60 frames correspond to a combined ~1.3 s of data surrounding the frame to be interpolated. This process did not alter qualitatively our main results (see Fig. 4g and Supplementary Fig. 8).

### Postprocessing pipeline

Our postprocessing pipelines were custom-written in MATLAB (Mathworks) and Python (www.python.org). Some routines utilized NumPy (https://numpy.org/) and Matplotlib (https://matplotlib.org/).

**Stimulus response measurement.** Unless stated otherwise, spike count over a window of 6 imaging frames (~132 ms) starting at the stimulation onset was used as a measure of the response to holographic stimulation of each target cell for all cells. That window was chosen because it includes the full stimulus duration (100 ms) and one extra frame after the stimulus offset, for which the increased activity in TC can still be observed (see Fig. 1f). This was compared to the baseline count, defined as a 6-frame window starting ~264 ms before stimulus onset. Due to difficulties and limitations of spike deconvolution of calcium traces, some parts of the time series resulted in artificially high spike rates. To avoid artificial increase of variability measures due to

these artifacts, we introduced an outlier removal procedure in which trials with spike count above 10 standard deviations over the mean are removed from the analysis. This procedure resulted in < 0.1% trials removed across all neurons for all experiments.

**Target selection and exclusion.** Up to 25 target cells were visually selected based on expression and location in the field of view (cells close to the edges were avoided) for each experiment. Not all targets were responsive to stimulation. Targets in which fewer than 20% of the trials evoked a spike count above the 91% percentile of its baseline count were excluded from the analysis. This procedure resulted in 54 target cells across 16 experiments from 6 mice being considered out of 193 targets attempted across 33 experiments from 11 mice. We estimate an average of $46.3 \pm 17.9$ ChrimsonR-labeled cells per field of view, which translates to an average density of labeled cells of $\sim 2.3 \times 10^{-2}$ cells per $100\ \mu m^2$. This sparse labeling facilitates the holographic stimulation of individual targets (see examples at Supplementary Fig. 1).

**Definition of significant responders.** Cells with significant response to a target's stimulation were defined as those whose spike count distribution was significantly different from its baseline count distribution. Specifically, we used a z-test to compare the stimulus response spike count to the average and standard deviation of the baseline count for each non-target cell. Unless stated otherwise, a cell was defined as a positive responder if the $z$ statistic is above the highest $\alpha = 2.5\%$ of the distribution, as a negative responder if the z statistic is below the lowest $\alpha = 2.5\%$ and as a non-significant responder otherwise. We have also corrected for false-discovery rate (FDR), by employing the matlab function mafdr[34,35], with an FDR of 2.5%. After FDR correction, the number of positive responders increased in relation to the one obtained at $\alpha = 2.5\%$, whereas the number of negative responders dropped to nearly zero. It is important to highlight that, given a low baseline activity rate for most cells, with an average count in the relevant window for our analysis near zero, it is technically challenging to define negative responders in this scenario using spike data. Other groups have shown suppressed neurons upon select pyramidal neuron stimulation using fluorescent traces, for which detection of significant decreases during stimulation might be easier[7,88].

This procedure is done for each target stimulated in an experiment separately. Therefore, in experiments where multiple targets were selected, each non-target cell is tested for the significance of its response to each target's stimulation, possibly being a significant responder for some targets and a non-significant responder to others (see Fig. 4e, inset).

**Fano Factor calculation and significance.** Fano Factor was calculated as usual, for each cell separately, by dividing the variance of their spike count by the mean of their spike count across trials. To correct for changes in firing rate across conditions[37], we introduced a procedure in which the window used to measure the spike count during stimulation varied from 1 to 9 frames ($\sim 22$ to 198 ms), starting at the onset. The window length was determined by minimizing the normalized difference between the mean baseline and stimulus counts. Cells for which this difference was above 20% were excluded from the analysis. This procedure resulted in less than 30% of cells removed, on average, across the non-target cells and an average of 65% of targets removed (minimum of 13% or 7 targets considered). Values were reported as mean ± error over the mean across cells, unless stated otherwise. Significance of Fano Factor changes across conditions was obtained by employing the Wilcoxon rank sum test at a significance level of 0.05 ($p$-values reported in the Results section and figure legends).

**Trial separation and controls.** To control for different conditions affecting the variability, the Fano Factor was recalculated and compared using a subset of trials. To control for target cell response variability affecting non-target cells, we separated trials into low target count trials and high target count trials. We defined low count trials as those for which the target cell produced a spike count during stimulation that was above the 91st percentile of the baseline count distribution and up to twice that number of spikes, whereas high count trials were any trial with a count above that limit. To control for the influence of ongoing fluctuation of the network activity we separated trials into low and high population baseline count, defined as trials for which the population count during baseline was lower than the 25th percentile of its distribution or higher than 75th percentile, respectively. To control for the temporal influence of the responses to the stimulus, we defined two consecutive 6-frame windows, centered at the peak of the target's average response to the stimulus ($2 - 3$ frames after onset), as early and late response windows.

**Correlation analysis.** Spike counts across trials were compared by computing correlations across time (autocorrelation, calculated using the spike counts over pairs of windows separated by a temporal lag of 132 ms) and across cells (comparing spike counts of targets vs. non-targets). Values are reported as mean ± error over the mean. Significance of correlation changes across conditions was obtained by employing the Wilcoxon rank sum test at a significance level of 0.05 ($p$-values reported in the Results section and figure legends).

## Decoding analysis

**Classification model selection, performance metrics, and data preprocessing.** In our prediction and classification scheme, we used target neuron ensembles as "classes" and the activity of the responding/non-target neurons as the "features". As described in the Stimulus response measurement section, we use spike count over a window of 6 imaging frames as a feature value for each neuron. For the temporal analysis, this window is offset relative to stimulus onset, as described below. In a preliminary analysis, we compared several machine learning strategies and found that the ensemble methods of boosting (XGBoost[38]) and bagging (scikit-learn[89] Random Forest classifier) performed best to several other common classification schemes such as Logistic Regression, Naive Bayes, Neural Networks, and Support Vector Machines. These two approaches yielded the highest accuracy and were then applied to all our analyses presented in this work. In all cases, we split the data samples randomly so that 80% was used for training and 20% for testing. We repeat this split 100 times and from these report the mean and standard errors of our metrics. As metrics, we use overall accuracy and per-class F1-scores (harmonic mean of recall and precision). For the results in Fig. 4g and Supplementary Fig. 10d, we use all data and all trials, but for others, we performed additional pre-processing before running classification, as detailed below.

Since not all trials were equally successful in eliciting spikes from target cells, for all other decoding analysis we selectively filtered trials and targets, with a criterion that the target neuron had to produce at least one spike across any of the six frames after the onset of stimulation. We then selected valid targets as those that spiked on at least half of their trials. Thus, non-significantly driven TC (based on conservative response spike count) excluded some TC in the spike count analysis which nevertheless had significantly F1-score in the decoding task (gray TC in Supplementary Fig. 10a). To characterize feature importances and interpret our classification schemes we used Shapley analysis[41], as described below.

**Decoding from baseline.** To examine whether TCs are decodable during spontaneous activity, we applied a similar analysis to the background activity in the periods between stimulation trials. We segmented this background activity using the same bin size (6 frames) as in the stimulation condition and identified pseudo-trials in which

only one of the previously used TCs spiked within a bin. To avoid potential artifacts introduced by Deep Interpolation, we excluded windows immediately preceding or following stimulation. For each of these background spike datasets, we restricted the analysis to the subset of TCs used during decoding in the stimulation condition and downsampled the data to match the sample distribution of the stimulation trials (Fig. 4c).

**Radius analysis.** For each target neuron, we implemented an exclusion zone based on the distance from the target neuron's soma, $r$. We progressively increased $r$, and the neurons/features falling within this distance were systematically eliminated from the feature matrix, enabling quantification of the relationship between F1-score degradation and exclusion zone radius for individual target neurons. We computed the probability distribution of any feature neuron being excluded, as a function of distance, $P_{ex}(r)$. Specifically, we aggregated the frequency of neuron eliminations at each radius across all datasets and normalized by the total number of exclusions.

The spatial dependency of predictive performance was characterized by analyzing the differential changes in F1-score across the range of exclusion radii. For each target neuron across all datasets, we identified the radius at which the decrease in F1-score first exceeded three standard deviations below the mean rate of change. These radii were averaged across datasets and smoothed using a Gaussian kernel ($\sigma = 1$) to obtain an estimate of the spatial scale of predictive influence.

**Neuron drop analysis and positive responder probability.** To evaluate the distributed nature of neural information and assess the robustness of our decoding framework, we conducted an iterative feature elimination analysis for each target neuron across all datasets. For each iteration, including initial model fitting, we use Shapley values (employed using the SHAP package: SHapley Additive exPlanations[90]) to quantitatively rank feature neurons based on their predictive contribution for a given target neuron. Through sequential elimination of the highest-ranked feature neurons and subsequent model retraining, we systematically assessed how the removal of informative neurons impacted decoding performance. This iterative process of SHAP-guided feature elimination and model reconstruction was repeated until a single neuron remained. At each iteration, we recorded F1-score, precision, and overall accuracy and the entire procedure was replicated one hundred times, utilizing different out-of-bag random selection of training and testing sets.

To characterize the relationship between feature importance and significant responders, we quantified the probability that eliminated neurons exhibited significant responses. For each target neuron, we constructed a binary vector corresponding to the sequential elimination order, where entries were assigned values of 1 or 0 based on whether the eliminated neuron was a significant responder. These binary sequences were averaged across all iterations of the elimination procedure to obtain target-specific probabilities and then averaged across all target neurons.

**Temporal profiles of the prediction accuracy.** To further characterize our holographic stimulation data, we explored the accuracy of our prediction scheme as a function of time. This procedure allowed us to study how long the information about the origin of stimulation persists in the network. Note that the prediction scheme is a more stringent requirement than studying only the causal effects of stimulation, since even though the activity after the stimulation can reverberate in a network for a long time, identifying the node that initiated it is more challenging. Furthermore, for the temporal analysis, we disabled our filtering of inactive targets/trials, utilizing all data. This simplifies the interpretation of these results, with an expected overall decrease in accuracy compared to the other decoding analyses.

We assessed the temporal course of prediction accuracy across datasets, each with different levels of responsiveness of the targets. We varied the size of our response window (1 or 6 frames) and then moved it in steps of 1. The main results are presented in Fig. 4g (single-frame analysis) using the data with $n = 10$ active targets and in Supplementary Fig. 10d (six-frame analysis) for all datasets. For the six-frame analysis, the fading "pink" rectangles indicate the amount of data used from the stimulation region (100 ms) window, so that the fully white regions use data strictly before or after stimulation is over. Note that the bins and the stimulation region are not perfectly aligned, as the trigger for the stimulation was independent from the image scanning raster. We define the reference bin 0 as the first bin which fully overlaps with the stimulation.

## Avalanche analysis

**Continuous epochs of suprathreshold population activity.** Continuous periods of population activity were identified by applying a threshold $\Theta$ on the population activity $p(t)$, the sum of the spikes from all neurons at a given time $t$, such that:

$$p_\Theta(t) = \begin{cases} p(t) - \Theta, & p(t) > \Theta \\ 0, & p(t) \leq \Theta \end{cases} \tag{4}$$

This procedure is known as soft-thresholding and was employed for all analysis unless otherwise stated. For a given recording $p(t)$ and coarse-graining value $k$, the dependence of the number $N$ of epochs on the threshold $\Theta$, $N(\Theta)$ was obtained for a range of thresholds $\Theta \in [\Theta_1, \Theta_2]$ such that $\Theta_1$ was low enough that it removed no population activity from the time course and $\Theta_2$ was high enough that it would remove all population activity from the time course. The function $N(\Theta)$ was typically well-approximated by a log-normal distribution, and a corresponding fit yielded shape parameters $\mu$ and $\sigma$. The threshold used in the analysis for all recordings and coarse-graining factors was chosen such that $\Theta = \mu$ and estimated for each $k$. See ref. 29 for details.

**Temporal coarse-graining.** A temporal coarse-graining operation was applied to the thresholded population activity $p_\Theta(t)$. For a given temporal coarse-graining factor $k$ an ensemble of $K$ different coarse-grained time series $p_k^0(\tau), p_k^1(\tau), \ldots, p_k^{K-1}(\tau)$ was arrived at through the following method:

$$p_k^j(\tau) = \sum_{i=k\tau+j}^{k(\tau+1)+j-1} p_\Theta(i) \text{ for } \tau \in \left[0, 1, \ldots, \left\lfloor \frac{T-j}{k} \right\rfloor\right] \tag{5}$$

For each time series $p_k^j(\tau)$, epochs were extracted by finding pairs $(\tau_1, \tau_2)$ such that $p_k^j(\tau_1) = 0$, $p_k^j(\tau_2) = 0$ and $p_k^j(\tau') > 0$ for all $\tau' \in [\tau_1 + 1, \ldots, \tau_2 - 1]$. The size of the epoch is given by

$$S = \sum_{i=\tau_1}^{\tau_2} p_k^j(i) \tag{6}$$

and its corresponding duration given by $\tau_2 - \tau_1 - 1$. For a given ensemble of coarse-grained time series, all epochs were pooled.

**Scaling curve fit.** For more precise evaluation of the scaling curve, mean avalanche size *vs.* duration, we introduced the following fitting function:

$$S(d) = \frac{Cd^{\chi_{sh}}}{\left(1 + (d/\Phi)^\gamma\right)^{(\chi_{sh} - \chi_{lg})/\gamma}} \tag{7}$$

This function is a double power law with initial slope $\chi_{sh}$, transitioning to a second slope $\chi_{lg}$ at around the point $d = \Phi$. The parameter $\gamma$ controls how abruptly that transition happens and has been fixed at 4 for

all the curves presented. The other parameters were free to adjust to the data, and all fits were performed in log-space, i.e., the $\log(S(\log(d)))$ was fit to the log of data (taking the log of both average sizes as well as durations).

## Simulations

We created a $300 \times 300$ two-dimensional lattice of excitable cellular automata[91]. We defined the spacing of the lattice as 12 µm, to mimic the high density of cells in brain tissue. We connected cells by a probability function that changes with the distance between cell pairs, with a short (gaussian, 100 µm characteristic decay) and a long (exponential, 290 µm characteristic decay) component

$$P_{conn} = 0.05 e^{-r^2/2 \times 100^2} + 0.03 e^{-r/290}. \qquad (8)$$

At each time step (set at 2 ms to mimic average spike transmission times in the brain), each cell can be in one of three states: (1) resting, (2) active, or (3) refractory. A resting cell can become active by either receiving an external Poisson drive (with probability $P_{poiss}$) or by an active presynaptic neighbor (with probability $P_{trans}$). We set $P_{trans} = \sigma/\bar{k}$, where $\bar{k}$ is the average number of connections per cell ($\sim$ 43 with the numbers described above) and $\sigma$ is the branching parameter, a variable we can adjust to move the system in and out of criticality and is expected to be slightly above 1 at the critical point (to compensate for energy loss/collisions in spike propagation in the network – $\sim$ 1.0066 on our simulations). We set $P_{poiss}(\sigma)$, so that the average amount of activity during baseline for each $\sigma$ value matches the average baseline activity in the experiments (see Fig. 7b). We note that for the supercritical networks (particularly $\sigma = 1.01$, see Fig. 6b bottom), the only way the average activity could be matched was by taking advantage of finite size effects, since in the supercritical phase there is self-sustained activity which persists even in the absence of drive. There is a non-zero probability of cascades of activity not being able to bootstrap themselves into this self-sustained activity level, given the size of the network and rate of drive. The pauses observed in Fig. 6b, bottom, are natural avalanche terminations that, given the extremely low Poisson drive applied here, result in long wait times until new activity is seeded by the external stimulus. Together, those average out to the same overall baseline activity rate observed in the other dynamical regimes. For this reason, a proper comparison of decoding accuracy of the supercritical network cannot be achieved (but see Fig. 7f and Supplementary Fig. 14b, c).

We then selected for each "experiment" (10 total), a non-overlapping set of 10 cells (i.e., 100 unique cells across the 10 experiments) in the center of the network (inner $38 \times 38$ cells) to be the target of extra external drive (as in the mice experiments, for 100 ms or 50 time steps, with 2 s inter-trial interval), which was set so the average activity of each target in the simulations during stimulation matches the average activity of targets during stimulation in the experimental data. All analyses are then performed exactly as done for the experimental data, but only using the inner $38 \times 38$ cells, to avoid border effects (periodic boundary conditions were used to simplify the simulations).

**Output strength.** To evaluate the relationship between the structure of the network and each target neuron's ability to spread information regarding their firing, we introduced the measure of output strength

$$OS(i) = \sum_{o=1}^{5} \sum_{j=1}^{N} (P_{trans}\mathbf{A})_{ij}^{o}, \qquad (9)$$

where $\mathbf{A}$ is the network's adjacency matrix ($A(i,j) = 1$ when there is a connection from neuron $i$ to neuron $j$, and 0 otherwise), $N$ is the number of neurons and $P_{trans}$ is the spike transmission probability. By taking $(P_{trans}\mathbf{A})$ to the power $o$, we evaluate the likelihood for a spike to transmit from $i$ to $j$ via an $o$-order path (e.g., order 1 would be a direct connection, order 2 would be a two-step connection – neuron $i$, to

neuron $k$, to neuron $j$; orders higher than 5 only negligibly contributed to $OS(i)$ calculation and were ignored). Therefore, after summing over all other neurons, $OS(i)$ estimates the number of spikes generated in the network for each spike neuron $i$ produced, excluding dynamics (collisions, refractory periods, etc.) and only considering the network's structure.

## Statistics & reproducibility

No statistical method was used to predetermine sample size. Mice/experiments without any responsive targets were excluded from the analyses ($n = 17$ experiments from $n = 5$ mice; see "Target selection and exclusion" section above). Experiments were not randomized, and the investigators were not blinded to allocation during experiments and outcome assessment. Animal sex was not considered in the study analysis due to the low number of animals included. For statistical analysis, please refer to each relevant Methods subsection for details on tests used and sample sizes. Schematic drawings and layouts (e.g., Fig. 1a and Supplementary Fig. 2a, d) were created using Adobe Illustrator (v28.7.8).

## Reporting summary

Further information on research design is available in the Nature Portfolio Reporting Summary linked to this article.

## Data availability

The pre-processed imaging data used in this study are available in the general repository Zenodo using the following link: https://doi.org/10.5281/zenodo.17834168. The source data for all figures and supplemental figures in this study are provided for this paper. Source data are provided in this paper.

## Code availability

Computer code used in this study is available at the following GitHub link: https://github.com/plenzd/NoveltyScaling.

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

## Acknowledgements

We thank Craig C. Stewart and members of the Plenz lab for help with animal surgery and care. We thank the members from our U19 BRAIN initiative grant for the many discussions and technical advice. This research was supported by the Division of the Intramural Research Program (DIRP) of the National Institute of Mental Health (NIMH), USA, ZIAMH002797, ZIAMH002971, and the BRAIN initiative Grant U19 NS107464-01. This research utilized the supercomputing resources of the National Institutes of Health (NIH, USA; Biowulf, http://hpc.nih.gov). The contributions of the authors were made as part of their official duties as NIH federal employees, are in compliance with agency policy requirements, and are considered Works of the United States Government. However, the findings and conclusions presented in this paper are those of the author(s) and do not necessarily reflect the views of the NIH or the U.S. Department of Health and Human Services.

## Author contributions

T.L.R. and D.P. conceived and planned the study; T.L.R., B.G., and V.S. performed experiments; A.V. took the lead in holographic setup design. T.L.R. took the lead in data analysis with support from B.G. and V.S. S.P. and R.S. took the lead in machine learning based decoding of experimental data. All authors contributed to the analyses. T.L.R. and D.P. wrote the manuscript.

## Funding

## Competing interests

The authors declare no competing interests.
