## [Transparent Peer Review file · Nature Communications]

Critical Scaling of Novelty in the Cortex

Corresponding Author: Dr Dietmar Plenz

Version 0:

Reviewer comments:

Reviewer #1

(Remarks to the Author)

This paper explores a fundamental question in neuroscience: how do novel, unexpected action potentials in a single cortical neuron influence the surrounding neuronal network, and what do these dynamics reveal about information processing in the brain? The authors employ a cool and novel all-optical approach wherein they combine two-photon calcium imaging with “holographic stimulation” to trigger spikes in individual layer 2/3 pyramidal neurons in the primary visual cortex of awake, resting mice. Simultaneously, the authors recorded the calcium activity of hundreds of surrounding neurons. Their findings reveal that when a single pyramidal neuron is stimulated, approximately 15% of surrounding neurons increase their calcium activity (positive responders), while fewer than 1% decrease their activity (negative responders). Critically, they observe that the network response scales with a power law relationship to the stimulation intensity, with an exponent of approximately 0.3 during stimulation compared to 0.5 during baseline activity. The authors demonstrate that this network can effectively decode which specific neuron was stimulated, achieving high accuracy despite substantial trial-to-trial variability. Furthermore, they show that these information processing dynamics occur within ongoing neuronal avalanches (which the authors interpret as a signature of critical dynamics). Based on these observations, the authors argue that criticality plays a fundamental role in enabling efficient communication of novel information across cortical networks.

While the authors argue that criticality plays a fundamental role in enabling efficient communication of novel information across cortical networks (and I suspect that they are 100% correct, philosophically), the study lacks necessary controls to substantiate this claim scientifically. The experimental design, though methodologically advanced, is purely observational with respect to criticality - it demonstrates that information processing occurs within a network exhibiting avalanches but makes no attempt to establish whether this processing would be impaired in non-critical states (or if criticality is unrelated to these phenomena). There are no experiments that push the system into supercritical or subcritical regimes to test whether the main findings, such as the power-law scaling relationship or high decoding accuracy, would dissipate. Without such perturbations, it remains plausible that the observed data could be explained by alternative mechanisms not requiring criticality, such as specific network connectivity patterns, intrinsic cellular properties, or perhaps a slightly supercritical system. This represents a significant limitation that undermines the paper's central claim. A more convincing approach would incorporate pharmacological or optogenetic manipulations known to disrupt critical dynamics, followed by measurements of how information processing changes in these altered states. Alternatively, the authors have a history of very rigorous modeling, and a similar set of evidenced-based priors could be established and tested in a computational context.

Major Concerns

1. There appears to be a contradiction in how the scaling exponent relates to critical behavior. The authors report a baseline exponent $h \approx 0.5$, which they note aligns with mean-field directed percolation (MF-DP) universality class. However, upon perturbation, they observe $h \approx 0.3$, which approaches 2D directed percolation. This shift in exponent values is presented as evidence for criticality, but the authors do not adequately address why this change occurs or what theoretical framework explains these specific values. If the cortical network truly operates at criticality, one would expect a consistent relationship to a single universality class. Without a theoretical model reconciling these different exponent values, it's difficult to interpret whether this finding supports or contradicts the criticality hypothesis.

2. The analysis of avalanche distributions relies on size versus duration relationships across a remarkably limited range. For example, in the $k = 9$ case, the power-law region spans only from 200 ms to 1 s - just one order of magnitude. Similarly, for $k = 1$, the range extends from 22 ms to approximately 300 ms. This restricted range raises serious questions about the

robustness of the power-law fits and the subsequent interpretations regarding critical dynamics. Standard practice in criticality studies typically requires observing power-law behavior across at least two orders of magnitude to make convincing claims about scale invariance. The authors should either provide stronger evidence spanning wider timescales or acknowledge this substantial limitation.

3. The study lacks essential controls that would strengthen claims about criticality. The authors assume criticality without testing alternative explanations for their observations. A slightly supercritical system might well account for these results, but this possibility remains unexplored. Two approaches could significantly enhance the work's validity: first, implementing perturbations that push the system into supercritical and subcritical regimes to test whether the main findings persist; second, developing computational models to mechanistically explain unexpected features of the data, such as why only 15% of neurons comprise the positive responder cohort. Given the authors' strong track record with modeling work, the absence of simulations that could test the criticality hypothesis is particularly notable.

4. The network architecture appears to play a crucial role in the observed phenomena, yet this aspect receives insufficient attention. The authors should examine whether pyramidal cells are uniquely positioned to produce scale-invariant responses or if other cell types would show similar behavior. Critical phenomena are typically associated with systems of codimension 2, suggesting the control parameter (related to neural connectivity) should be manipulable to demonstrate a break in critical behavior. The study would benefit from investigating how disrupting the connectivity of target cells affects scaling properties. Additionally, a functional network analysis could reveal meaningful information about the topology and connectivity patterns that support the observed dynamics - information that is typically missed by using encoding-decoding machine learning algorithms alone.

5. There seems to be a disconnect between the current findings and previous work on criticality and dynamic range. Prior research has shown that dynamic range is maximized at criticality (Shew et al. 2009), yet despite observing scale invariance in the response, the range in this study appears limited. If the network truly operates at criticality, one would expect the response to perturbation to propagate extensively, potentially reaching the boundaries of the system. The authors should reconcile these seemingly contradictory observations and clarify why the propagation of novel information appears more constrained than theoretical predictions would suggest in a truly critical system.

Minor Concerns

1. The manuscript would benefit from clearer visualization and presentation of the spatial distribution of responding neurons. While the authors report that positive responders (PosR) are distributed throughout the network, the figures do not adequately convey the spatial patterns of activation or the relationship between distance from the target cell and response magnitude. A more comprehensive spatial analysis would help readers understand the propagation dynamics.

2. The authors use Deep Interpolation for denoising their calcium imaging data, which improves signal quality but may also introduce temporal artifacts. The supplementary data suggests this processing slightly advances stimulus information and prolongs transient response time. The manuscript should more thoroughly address how this preprocessing might influence the temporal dynamics of the observed responses and potentially affect the avalanche analysis.

3. The distinction between genuine network responses and potential direct optical stimulation of non-target cells requires further clarification. While the authors performed a radius exclusion analysis as a control, more detailed characterization of the holographic stimulation profile would strengthen confidence in the specificity of their approach.

4. The paper lacks a clear discussion of how the observed scaling properties might relate to behaviorally relevant information processing. While the authors demonstrate that network responses encode information about which neuron was stimulated, they do not connect these findings to functional or computational advantages for the animal. This connection would enhance the broader impact of the work.

5. Is the statistical approach used to identify positive/negative responders sensitive to the specific threshold? The authors might consider demonstrating robustness across a range of statistical thresholds, or ideally, adopt a more continuous measure of responsiveness that doesn't rely on binary classification.

Reviewer #2

(Remarks to the Author)

This work presents an investigation of the resulting effects on the surrounding network of perturbing the firing of a target neuron in mouse cortex. The authors use optogenetic techniques to stimulate and record the activity of the neurons. The results are substantial and novel, generating interesting takeaways and ideas for future exploration.

Most of our feedback is relatively minor and offered to modestly increase the impact of the work with what we believe should be relatively little effort.

Noteworthy results:

1. Injecting spikes into the cortical network causes some neurons to increase their firing rates without decreasing the Fano Factor (as would be expected when driving the network with sensory stimuli). A small population of neurons decrease their firing rates as a result of the injected spikes and some neurons do not significantly change their firing rates.

2. A substantial portion of the neurons in the network respond by increasing their firing rates when the target neuron (TC) spikes, a higher proportion than neurons that are direct postsynaptic targets of the TC neuron. This is the case whether the target neuron spikes intrinsically (baseline) or spikes in response to optogenetic inputs (response).
 3. The network activity following intrinsic spikes (baseline) is significantly different from the response to optogenetically-triggered spikes (response).
 - A. Populations that do not respond to the optogenetically-triggered TC spikes (NonR) increase spiking in concert with intrinsic TC spikes (this may be caused by both TC neurons and NonR neurons being driven by some third factor, as opposed to the TC neurons driving the NonR neurons).
 - B. Populations that correlate negatively to the optogenetically-triggered TC spikes correlate positively to intrinsic TC spikes.
 - C. Neurons that respond positively to optogenetically-triggered TC spikes (PosR) increase their firing rate as the firing rate of the TC neuron increases. However, the increase is actually at a faster rate as a function of intrinsic TC spikes than of novel TC spikes.
- In particular, point 3C above to us suggests that there may be an underlying "neural manifold", and that optogenetically-triggered TC spikes are "off-manifold" perturbations, and the circuit limits the extent to which these off-manifold spikes drive activity, similar to other studies.
4. The identity of which neuron was the optogenetically-driven TC neuron can be reliably decoded from the population activity, but this decodability vanishes after about 200 ms.
 5. Decodability is more or less robust to changes in the underlying dynamical state of the network, as measured by being in a Low firing or High firing avalanche state.

Novelty and impact

Previously the impact of injecting a single spike into a cortical circuit has been investigated in London et al. (reference 2). The above results significantly extend this in interesting ways that shed substantial conceptual light on the circuit.

Support for results/claims

The presented data does appear to generally support the major claims made in the paper, although we believe additional clarification should be made in terms of the expectations set by previous models, experiments, and basic reasoning. We also make suggestions for additional analysis that we believe would strengthen the support for the major claims.

1. The authors state that the scalings observed in Figs. 2d,e support "efficient" coding. Could you clarify which null model or baseline is used to claim the efficiency? Additionally, please explain how having an exponent that is either too large or too small might be detrimental. Is the 0.5 degree scaling with the intrinsic TC spiking more or less efficient than the 0.3 degree spiking of the induced TC spikes?
2. Related to the preceding point 1, the authors show in Figure 2 critical exponent scaling that they claim should be approximately optimal for propagating information, and in Figure 3 they measure information propagation, but they do not connect the two directly by, say, linking higher or lower critical exponents to more or less decodability.
3. Related to the preceding point 2, we believe it would strengthen the results to include a version of Fig. 3i for the case of decoding the identity of a TC neuron that emitted a baseline/intrinsic spike. This would help to deduce if the information spreading through the network as a result of the novel spikes spreads more or less efficiently than the information spreading as a result of intrinsic/non-novel spiking.
4. We believe that the results of the paper would be strengthened by showing the scaling exponents of Fig. 2d,e separately in the Low and High firing regimes of Fig. 4f. Our reason for this is that we feel that Fig 4 attempts to show how the main important results of the preceding figures depend on the global network state, and these scaling exponents appear to be central to the message of the paper alongside the decodability.
5. This point concerns the claims about the network being an efficient propagator of information. A possible alternative interpretation for Fig 2d versus 2e, and Fig. 4i, might be that the network is actually good at quashing the amplitude information of perturbations (as opposed to enhancing these perturbations), as stated on line 126 "network responses to single spikes scale similar to responses to multiple novel spikes from a single neuron." From this perspective, the injected spikes might be better understood as noise to be suppressed rather than as a signal to be amplified. Again, having more explicit baseline models or expectations in mind may help contextualize the results.

Soundness and thoroughness of methodology and analysis

While the primary reviewer is not very familiar with the type of methodology used in this study, to the best of their ability it appears to be sound and sufficiently described to be reproducible.

Minor comments

We believe that equations 1 and 2 would be clarified by writing the dependence of the variables on baseline/stimulation, or by writing (baseline/stimulation) on the same lines as the equations. This may want to be carried out throughout the paper, for instance on line 135, where writing that TC and PosR are here the stimulation TC and PosR variables would clarify things significantly.

When measuring spike scaling, the authors must determine the number of spikes emitted by the target neuron (TC). The authors chose a 132ms window over which to count these spikes. It would be nice to have some justification for this choice of interval.

The authors use a threshold α to mark populations of neurons which are impacted by the target neuron perturbation, but don't appear to define this in the Methods. α should also be defined first on line 97, and it should be made consistent whether "alpha" or " α " is written in the text.

We believe that a more explicit discussion of the advantages of the holographic technique, compared to the results reported in reference 2 (London et al.), would be valuable.

Figure 2d appears to be missing a legend. It may be sufficient to state in the caption for Figure 2d that the legend can be found in 2e.

The authors frequently state that the power-law scaling is "robust" -- does this refer exclusively to the robustness with respect to alpha as shown in Fig 2g, or is it meant to be taken in a more general sense. If it is more general, what is the justification for this?

What is the definition of $\langle cc \rangle$ just before equation (1)?

It would aid the reader to clarify that in the caption for Figure 2 the SD is across neurons. This clarification could be done in the Methods.

We believe it would be clearer if the lightning bolt in Fig 1c points to the lefthand part of the appropriate interval.

Only fewer than 1% of neurons showed negative responses to TC stimulation. Are they genuinely negatively responding or found by chance? It is helpful to clarify how the replicability analysis was conducted in the context of multiple statistical testing.

Reviewer #3

(Remarks to the Author)

Reviewer #4

(Remarks to the Author)

Reviewer #5

(Remarks to the Author)

Ribeiro et al. present a technically impressive study using advanced all-optical stimulation and recording methods to examine how cortical networks respond to novel stimuli. This work continues a relatively recent line of investigation into the coding schemes employed by cortical circuits. As outlined effectively in the discussion, a central challenge in the field is bridging the gap between what is known about the integrative properties of single neurons and the connectivity/activity patterns observed at the network level, and the apparent sensitivity of these networks to small, unexpected inputs.

According to standard synaptic integration models, such small perturbations should exert minimal influence on overall network activity. However, a growing body of experimental work has demonstrated that such perturbations can have amplified, long-lasting effects, even influencing perception and behavior. Understanding this phenomenon holds significant implications for our understanding of neural coding and cortical integration of input.

In this context, both the premise of the manuscript and the methodology used are highly promising. The experiments themselves are technically demanding, and the authors deserve credit for executing them successfully. Moreover, the core finding - that perturbations of as few as 1-7 neurons can be reliably detected at the population level provides compelling evidence that this form of sensitivity is indeed a robust property of cortical networks.

However, the manuscript as a whole is difficult to follow, for several reasons:

1. Conceptual and Terminological Obscurity: The writing lacks clarity and makes liberal use of highly technical terms (e.g., "criticality," "parabolic neuronal avalanches") without adequate explanation or motivation.
2. Inadequate Methodological Transparency: There is a lack of raw data presentation, and many key analytic steps are described only superficially. For instance, much of the analysis is done at the spike level, yet no demonstration is provided of how spikes were extracted from the calcium imaging signals.
3. Unclear Reference Framework: Some of the results show clear effects, but the lack of a well-defined null model or comparative baseline (e.g., a balanced network with mild external drive) makes it difficult to interpret their significance. At times, I was left unsure whether the paper tackles deep and subtle questions that are simply hard to understand, or whether it assembles a set of loosely connected findings without a unifying conceptual framework.

Given these issues, I would recommend that the authors be invited to revise the manuscript, possibly reusing the same data, but with a more transparent, accessible presentation of methods and results. The analysis methods should be explained in greater detail, assuming less prior knowledge from the reader regarding complex neural dynamics.

Specific Comments:

1. Abstract: The abstract is difficult to parse. For instance, "Their influence scales robustly to an exponent between 0.2 and 0.3 relative to their number" is opaque. Similarly, the phrase "This heightened susceptibility... aligns with the behavior of complex systems exhibiting critical dynamics".
2. Lines 144–148: The discussion of statistical thresholds (α) and PosR identification lacks rigor. Simply lowering the statistical threshold increases the number of detected "responsive" cells, but no control is provided to account for the increased false positive rate. This weakens the argument that the main finding is robust.
3. Lines 156 onward: The Fano Factor analysis is presented without sufficient methodological detail. It's unclear for which population of cells it was computed (e.g., PosR with default α or those under more permissive criteria). The conclusion — that a single novel spike elicits large-scale responses — seems overstated given the uncertainty in the permissive response

criteria. Including examples of raw data (e.g., as in O’Rawe et al.) would greatly enhance credibility.

4. Figure 3: The decoding of stimulation targets using population activity is interesting, but the underlying assumptions and statistical controls of the decoding procedure are not clearly explained. In large recurrent networks operating near criticality, the ability to back-infer the source of stimulation might depend strongly on hidden variables (e.g., direct synaptic links or network topology), which are not discussed.

5. Avalanche Analysis: The treatment of avalanche dynamics and parabolic scaling is not adequately explained in the main text. I found it very difficult to follow both the rationale and implementation of the analysis — even after reading the methods.

Minor Comments:

- Figure 1b (top): It is unclear what is being plotted. Is this the z-score of the target neuron? This should be made explicit.
- Line 119: The definition of TC_{sc}^h is unclear. It appears to mean $(\text{TC}_{sc})^h$, but this should be explicitly clarified.
- Line 127–130: The discussion in these lines is hard to follow. Rephrasing for clarity is needed.
- Line 133: Were these trials excluded from the analysis?
- Line 146: The phrasing regarding α is ambiguous. “Lowering α ” usually means increasing stringency, but here it seems to imply using a more permissive threshold. Please clarify.

Reviewer #6

(Remarks to the Author)

Ribeiro et al. aim to characterise the network effects of novel action potential activity in the primary visual cortex, linking this to a potential role in the detection of stimulus novelty. To explore this, they leverage holographically targeted optogenetics, aimed at the stimulation of a single neuron, during spontaneous activity in the visual cortex. They proceed to analyse the effect of stimulation on that neuron, and the changes in activity patterns in the population of simultaneously recorded neurons in the local network. This allows the authors to infer potential functional connectivity within the network and to look at variation in the stimulation driven modulation of network activity in the context of baseline activity levels.

To the best of my knowledge the fact that they are carrying out these manipulations while the animal is not experiencing salient visual stimulus, separates the work from previously published studies reporting similar experiments (Chettih and Harvey, 2019, Russell et al. 2024). It would seem however to somewhat limit the strength of connection they are attempting to supply, between their results and the detection of novelty in real world sensory stimulation, as the brain state will not be comparable.

The methods employed are sound and cutting edge and the results are potentially high impact. I do however have some significant concerns and suggestions for improving the manuscript.

Major

1. The major concern I have is the potential presence of off-target stimulation and the effect this could have on the interpretation of the data. The opsin they are using is mainly soma restricted, however there are still likely to be some off-target effects, and I would like to see this properly characterized. I have not seen this particular method of targeting used before which further adds to the need for resolution quantification.

a. The authors should create physiological resolution plots by varying the location of the stimulation beam relative to the centre of the targeted cells soma (e.g. Marshel et al. 2019). This should be done across X, Y and Z, planes. This should be achievable with the hardware and software they have used. The resolution curve produced should then be used to determine the spatial cut of threshold for excluding any ROIS from their analysis. It is clear from Fig. 2h that the majority of positive followers are in proximity to the target (0~75 microns)

b. The Z plane will be the most limited resolution in this setup and is likely to stimulate unobserved neurons above and below the target cell. This should be mentioned in the discussion.

c. The GG to GR alignment and the Top-Hat profiles show in supplemental figure 2 are not perfect which furthers the need for this quantification. It would be good to show a larger area on the examples in supplemental figure 3 to better assess off-target stimulation.

d. In Supplemental figure 3 there are clear examples of stimulation more than one neuron, for example target 27 and 45. There are also large areas of positive fluorescence change around the targets, I am curious how the authors explain this when the net surround has been shown to be inhibitory previously (e.g. Russell et al. 2024). Is it uncorrected neuropil? Off-target activation of dendrites? Artifact from the stimulation laser?

e. The magnitude of the response they report in ‘follower’ neurons looks relatively big compared to previous reports (Fig 1c for example). This increases the concern over any off-target stimulation effects. I am curious why the authors think they see such strong responses resulting from a few spikes in a single neuron. Perhaps it is partially due to the brain state they are studying, this should be discussed more in the manuscript and compared to previous data.

f. In addition to the resolution quantification, it would be beneficial to display the average movement they are seeing in their preparation. This is relatively easy to do using the values of the motion correction performed in suite2p.

g. One of the arguments that they are not seeing off-targets near target neurons is that they don’t see a decrease in decoder performance when removing cells within 10 microns of the target. Is this taken from the centroid of the target neuron to the centroid of the non-targeted one? If so how many cells actually fall within this range? It is possible that the lack of decrease in decoder performance is due to a lack of neurons in this area.

2. In Fig. 1a the authors show, and indeed label, the presence of clear activation of putative dendrites in the imaged FOV.

These ROIs are fairly large and can be a large distance from the soma (one is approximately 80 microns away in this example). It is likely that these are part of the target neurons and it is imperative that these ROIs are removed from all analysis. They would produce ROIs with a very high magnitude response and will ensure the possibility of decoding the targeted neuron from the other observed activity. These ROIs need to be removed for the data to be properly interpreted. They could be filtered out using the footprint of the ROI, then potentially the unusually high magnitude of response and extremely high correlation in activity to the target soma. Success should be confirmed manually.

3. Artifact from the stimulation laser is a serious concern with such experiments, particularly when relying on small difference in fluorescence traces for readout. Hopefully there is no issue with the amount of total power being used staying relatively low. It would however be good to confirm this by looking at the time of stimulus onset across all ROIs and looking at the raw fluorescence traces for any sudden increase. If a notable deflection is observed, then the authors may need to attempt artifact subtraction.

4. The data is heavily processed which makes it harder to interpret accurately. As a sanity check I would be grateful to see some examples of the same analysis but being conducted using the DF/F rather than the inferred spikes (for example Fig. The authors mention themselves that this process can cause issues, and they have had to remove a small amount of data. Does this impact the number of modulated neurons observed?

5. The analysis to identify modulated neurons could benefit from correction for false discovery rate, this may reduce the somewhat surprising number of neurons that are being identified as modulated (e.g. Fisek et al. 2023).

6. I find the plots looking at the transient nature of the decoder performance around stimulation in appropriate for the purpose. Why are they produced in such a way as to give the large increase in performance around the stimulation? (both before and after). How would the decoder possible function before the stimulation of the target cell? In one session they report an increase in performance beginning >200ms prior to the stimulation onset. It would be insightful to see these plots made from rawer data, DF/F rather than inferred spiking, and without the large sliding window they have employed which ruins the temporal resolution.

Minor

It would be good to clarify somewhere that it is more difficult to detect inhibitory effects as these require a significant baseline activity and tend to be lower in magnitude. This relates to the discussion of the brain state, with different levels of activity and inhibition.

Figure 2b and others – I can't really distinguish the bars with or without the patterning in a printout of the manuscript. The authors should consider making this easier to distinguish.

Supplemental figure 3 would benefit from a scale bar and the distance being given in μm rather than pixels in the legend.

Line 47 – I am not sure that 'Optogenetical' is a word, perhaps optogenetic.

Line 170 – 'Network' needs an s on the end

Version 1:

Reviewer comments:

Reviewer #1

(Remarks to the Author)

We have carefully reviewed the revised manuscript and the authors' responses to our concerns. The authors have made substantial improvements to the work, particularly through the addition of rigorous computational simulations that provide important theoretical grounding for their experimental findings. These simulations effectively demonstrate that the observed experimental results align with network dynamics operating near a critical state, significantly strengthening the manuscript's central claims about cortical information processing.

While we acknowledge that some theoretical inconsistencies remain unresolved—specifically the apparent contradiction between obtaining 2D directed percolation scaling exponents from perturbation responses while observing mean-field directed percolation characteristics in avalanche distributions—the authors have provided reasonable explanations for the limitations inherent in two-photon imaging approaches. Their acknowledgment that baseline versus perturbation comparisons are inherently difficult due to correlated ongoing activity is appropriate, and their argument that the power-law response curves themselves provide evidence of criticality is acceptable within the broader context of their findings.

The experimental work remains technically sophisticated and methodologically sound. The core findings regarding single-neuron perturbation effects and their propagation through cortical networks represent novel and significant contributions to our understanding of cortical coding mechanisms. The demonstration that individual action potentials can reliably encode information about their origin across distributed networks, despite high variability and ongoing avalanche dynamics, provides important insights into cortical information processing.

The authors have adequately addressed our concerns about experimental controls, methodological transparency, and the need for theoretical framework. The addition of simulations particularly strengthens the work by providing mechanistic explanations for the observed phenomena and demonstrating the relationship between network criticality and information transmission efficiency.

We recommend acceptance of this manuscript. The work makes meaningful contributions to neuroscience despite remaining theoretical complexities, and the experimental and computational findings will be of significant interest to the broader community studying cortical dynamics and neural coding.

Reviewer #2

(Remarks to the Author)

We thank the reviewers for their thoughtful and substantial response and revision. We believe that the addition of the accuracy plot in Figure 4b is very interesting and really contributes to the investigation of “novel scaling of criticality”, since we feel that an investigation of novel spikes should compare against non-novel spikes. The addition of Figure 6 also helps to provide a connection between the 0.285 scaling observed in experiments and purported criticality, and to establish the functional significance of this criticality (encoding novel inputs in a way that can be more successfully decoded).

Regarding the new Figure 6, we do have some reservations about how universal this correspondence between the 0.285 exponent scaling and criticality is among appropriate networks for modeling cortex, especially since even very small deviations appear to correspond with networks outside of the critical regime (inset of Fig. 6f). In addition, Supplementary Figure 11c indicates that the scaling relationship breaks down in the higher firing rate regime, unlike what is experimentally observed in 11a. If the network simulations require fine-tuning, we feel that this should be pointed out, and in general the robustness to changes in model parameters of the 0.285 exponent scaling at criticality should be discussed.

In addition, we believe that there is still one more piece remaining to fully connect novelty, scaling, criticality, and functional significance (novel stimulus decodability). As in the new accuracy plot for Figure 4b, we believe that the key comparison between the base and response cases should be extended to Figure 6, in order to show the effect of novel spikes in the simulated model. We believe this would entail the following: a version of Fig. 6c where the abscissa is the firing rate of non-TC neurons; a version of Fig. 6e where the target count is replaced by the number of non-TC spikes; a version of Fig. 6f where the slope gamma is measured for the non-TC spikes; and most importantly, versions of 6g and 6h where the accuracy and F1-scores are measured when decoding non-TC spikes. Taken together, this should show that novel events (as opposed to non-novel events) are best encoded by critical networks, and that appropriate network models of the cortex generically exhibit ~ 0.285 exponent scaling at criticality.

We think it would also be beneficial to extend Figures 6f, g, and h to sigma values in the supercritical regime, and to add two more sigma values in Figure 6e to show (nearly) flat lines more fully in the supercritical and subcritical regimes (perhaps sigma values of 0.9 and 1.01 to correspond to those chosen elsewhere). Please add to the model description the mechanism that shuts down the bursting activity in the supercritical network after approximately eight seconds in Fig. 6b.

It seems to us that the question of mechanistically why novel spikes are more decodable than non-novel spikes would be an interesting investigation. The authors may want to add such a question and insights from the model to the Discussion.

In Fig 6c and 6e, the dashed lines should be clearly defined in the figure caption.

Reviewer #3

(Remarks to the Author)

Reviewer #4

(Remarks to the Author)

Reviewer #5

(Remarks to the Author)

First, thank you for the very substantial revision. Responding coherently to six reviewers is no small feat; the manuscript is much clearer and tighter as a result. Almost all of my original points were addressed satisfactorily.

However, I would like to highlight one point that may not have been explicitly enough addressed in my previous report and that still remains unaddressed. In my last report, I noted that the absence of reference and null models makes it difficult to evaluate some of the results. Specifically, I want to focus on the avalanche analysis. While the exposition and parabolic shape results are clearer now (Fig. 5b–d, f–h), the claim would be stronger if you included a surrogate/permutation analysis. This would help demonstrate that scaling is not an artifact of thresholding, finite windows, or non-stationarity. I suggest running the same analysis pipeline on null models that preserve low-order statistics but break the cascade structure. For example, you could try one of the following: Circular time-shift of the stimulation train, or a Trial/epoch block-shuffle of it, or an event-time jitter ($\pm 1, 2$ bins). I'd expect that the power-law size/duration would weaken or fail under these nulls, which would strengthen your findings.

Additionally, it would be helpful to include a one-line limitation statement in the discussion. It should note that because the avalanche analyses are computed within finite imaging epochs and fields of view, the duration and size distributions are truncated and can be biased by these finite-window effects. This would help remind readers that the exponents should therefore be interpreted with caution.

Reviewer #6

(Remarks to the Author)

Thank you for addressing the comments and for all the hard work.

Version 2:

Reviewer comments:

Reviewer #2

(Remarks to the Author)

We thank the authors for their revision. We believe that our concerns were adequately addressed and just have a few minor points for improving clarity.

In Figure 4b, please specify which bar corresponds to accuracy and which to F1 score.

In Figure S13a, please write in or state in the caption the slope value for the dashed line.

In Figure S14b, please define what threshold is more explicitly.

In Figures S6g and 14b, please specify that accuracy is with respect to decoding the TC spike.

We agree that it would be interesting for future work to explore using inhibition, such as a negative Poisson drive, in the supercritical regime. As is we are satisfied with the approach taken here for the scope of this work.

Reviewer #3

(Remarks to the Author)

Reviewer #5

(Remarks to the Author)

The authors have answered all my concerns appropriately. I recommend publication. Congratulations on a remarkable work that combines state-of-the-art experimental and theoretical methods. I still find it challenging to wrap my head around the decoding findings. I hope it will steer enough attention to encourage further studies into it.

REVIEWER COMMENTS

Reviewer #1 (Remarks to the Author):

This paper explores a fundamental question in neuroscience: how do novel, unexpected action potentials in a single cortical neuron influence the surrounding neuronal network, and what do these dynamics reveal about information processing in the brain? The authors employ a cool and novel all-optical approach wherein they combine two-photon calcium imaging with “holographic stimulation” to trigger spikes in individual layer 2/3 pyramidal neurons in the primary visual cortex of awake, resting mice. Simultaneously, the authors recorded the calcium activity of hundreds of surrounding neurons. Their findings reveal that when a single pyramidal neuron is stimulated, approximately 15% of surrounding neurons increase their calcium activity (positive responders), while fewer than 1% decrease their activity (negative responders). Critically, they observe that the network response scales with a power law relationship to the stimulation intensity, with an exponent of approximately 0.3 during stimulation compared to 0.5 during baseline activity. The authors demonstrate that this network can effectively decode which specific neuron was stimulated, achieving high accuracy despite substantial trial-to-trial variability. Furthermore, they show that these information processing dynamics occur within ongoing neuronal avalanches (which the authors interpret as a signature of critical dynamics). Based on these observations, the authors argue that criticality plays a fundamental role in enabling efficient communication of novel information across cortical networks.

While the authors argue that criticality plays a fundamental role in enabling efficient communication of novel information across cortical networks (and I suspect that they are 100% correct, philosophically), the study lacks necessary controls to substantiate this claim scientifically. The experimental design, though methodologically advanced, is purely observational with respect to criticality - it demonstrates that information processing occurs within a network exhibiting avalanches but makes no attempt to establish whether this processing would be impaired in non-critical states (or if criticality is unrelated to these phenomena). There are no experiments that push the system into supercritical or subcritical regimes to test whether the main findings, such as the power-law scaling relationship or high decoding accuracy, would dissipate. Without such perturbations, it remains plausible that the observed data could be explained by alternative mechanisms not requiring criticality, such as specific network connectivity patterns, intrinsic cellular properties, or perhaps a slightly supercritical system. This represents a significant limitation that undermines the paper's central claim. A more convincing approach would incorporate pharmacological or optogenetic manipulations known to disrupt critical dynamics, followed by measurements of how information processing changes in these altered states. Alternatively, the authors have a history of very rigorous modeling, and a similar set of evidenced-based priors could be established and tested in a computational context.

Response: We appreciate this reviewer's support of our experimental findings and encouragement to strengthen our criticality claims. In this revision, besides additional analyses, we now include rigorous modeling which links our experimental results to insights from critical network dynamics as detailed below.

Major Concerns

1. There appears to be a contradiction in how the scaling exponent relates to critical behavior. The authors report a baseline exponent $h \cong 0.5$, which they note aligns with mean-field directed percolation (MF-DP) universality class. However, upon perturbation, they observe $h \cong 0.3$, which approaches 2D directed percolation. This shift in exponent values is presented as evidence for criticality, but the authors do not adequately address why this change occurs or what theoretical framework explains these specific values. If the cortical network truly operates at criticality, one would expect a consistent relationship to a single universality class. Without a theoretical model reconciling these different exponent values, it's difficult to interpret whether this finding supports or contradicts the criticality hypothesis.

Response: We now clarify that

- 1) *The expected behavior outside criticality would be either linear scaling (subcritical) or no scaling (supercritical) and therefore any sublinear response is of importance (last subsection on Results, 4th paragraph);*
- 2) *During baseline, the observed scaling is not a response of the network to the target neuron's firing, since during ongoing activity that firing is intrinsically and highly correlated with the rest of the network. These differences make comparisons (exponent during baseline vs exponent during perturbation) difficult if not impossible (Discussion, 5th paragraph).*

2. The analysis of avalanche distributions relies on size versus duration relationships across a remarkably limited range. For example, in the $k = 9$ case, the power-law region spans only from 200 ms to 1 s - just one order of magnitude. Similarly, for $k = 1$, the range extends from 22 ms to approximately 300 ms. This restricted range raises serious questions about the robustness of the power-law fits and the subsequent interpretations regarding critical dynamics. Standard practice in criticality studies typically requires observing power-law behavior across at least two orders of magnitude to make convincing claims about scale invariance. The authors should either provide stronger evidence spanning wider timescales or acknowledge this substantial limitation.

Response: We acknowledge this limitation, which is present on most reports of neuronal avalanches obtained from 2-photon imaging methods. We refer to the Capek et al. 2023 paper that goes into more details on why this limitation is there and why it should not prevent the criticality interpretation. In short, the limitation is introduced by our small window of observation into the brain. See also Miller et al. 2019, where we can see the scaling over a longer range in time when measuring it on a larger spatial scale (monkey LFP). We also would like to point out that the power-law response curves themselves provide evidence of criticality for our datasets.

3. The study lacks essential controls that would strengthen claims about criticality. The authors assume criticality without testing alternative explanations for their observations. A slightly supercritical system might well account for these results, but this possibility remains unexplored. Two approaches could significantly enhance the work's validity: first, implementing perturbations that push the system into supercritical and subcritical regimes to test whether the main findings persist; second, developing computational models to mechanistically explain unexpected features

of the data, such as why only 15% of neurons comprise the positive responder cohort. Given the authors' strong track record with modeling work, the absence of simulations that could test the criticality hypothesis is particularly notable.

Response: We have added rigorous simulations to the manuscript, with finite-site recording conditions in a random network with local spatial connectivity. We have added two new main figures (Figs. 6, 7) that examine expectation of novel spikes in subcritical, critical, and supercritical conditions. We demonstrate that our experimental results are in line with those coming from a system near criticality, including positive responder numbers, response curves, target decoding, and neuron dropping.

4. The network architecture appears to play a crucial role in the observed phenomena, yet this aspect receives insufficient attention. The authors should examine whether pyramidal cells are uniquely positioned to produce scale-invariant responses or if other cell types would show similar behavior. Critical phenomena are typically associated with systems of codimension 2, suggesting the control parameter (related to neural connectivity) should be manipulable to demonstrate a break in critical behavior. The study would benefit from investigating how disrupting the connectivity of target cells affects scaling properties. Additionally, a functional network analysis could reveal meaningful information about the topology and connectivity patterns that support the observed dynamics - information that is typically missed by using encoding-decoding machine learning algorithms alone.

Response: We agree that understanding differences in scaling behavior between cell types (e.g. pyramidal cells vs. interneurons) would be highly desirable to elucidate the role of inhibition in these dynamics. Unfortunately, currently available 2PI indicators don't work well to measure interneuron activity and are thus outside experimental reach.

However, we have extended our functional connectivity analysis and prediction in our simulations to emphasize the important point by this reviewer on how topology/architecture reflects functionality.

In our initial submission, we reported that the cross correlation found between PosR and TC during ongoing activity (base) does not predict the correlation between those cells during stimulation (detailed more clearly in our new Supplementary Fig. 7). In addition, we now show in our simulations that

1) the number of positive responder scales with a target cells' output strength, a measure of overall connectivity (on various degrees) of a neuron (new Fig. 7c),

2) the fraction of neurons that are positive responders decays with distance from TC in a qualitatively similar way as in the experimental data, with ~93% of neurons with a direct input from TCs being classified as PosR. However, only 16% of PosR had a direct connection from TCs, supporting the idea that the network of PosR of a given TC relies on the functional connectivity, beyond simple, direct synapses.

5. There seems to be a disconnect between the current findings and previous work on criticality and dynamic range. Prior research has shown that dynamic range is maximized at criticality (Shew et al.

2009), yet despite observing scale invariance in the response, the range in this study appears limited. If the network truly operates at criticality, one would expect the response to perturbation to propagate extensively, potentially reaching the boundaries of the system. The authors should reconcile these seemingly contradictory observations and clarify why the propagation of novel information appears more constrained than theoretical predictions would suggest in a truly critical system.

Response: We thank the reviewer for this question. We want to clarify that: 1) yes, the dynamic range is supposed to be maximized at criticality, but it is not supposed to be boundless (we are dealing with finite systems); we did not make claims on the extent of this response; 2) we do see responses that span almost the whole field of view, as we lower our criteria for significance of responders. Therefore, we do not believe there is a contradiction here. Also, in the brain, inhibition would presumably prevent small perturbations to grow to the boundaries of the system, as this would most certainly be detrimental to its information processing capacities.

Minor Concerns

1. The manuscript would benefit from clearer visualization and presentation of the spatial distribution of responding neurons. While the authors report that positive responders (PosR) are distributed throughout the network, the figures do not adequately convey the spatial patterns of activation or the relationship between distance from the target cell and response magnitude. A more comprehensive spatial analysis would help readers understand the propagation dynamics.

Response: In our new figure (Supplementary Fig. 4), we have added several examples of target cells and their significant responders' spatial distribution with cellular resolution detail. The summary of the distance dependence between targets and significant responders can now be found in Fig. 2h (see also new Fig. 7b on simulation results). We note that our temporal resolution of ~22 ms prevents us from identifying precisely the propagation dynamics in time and space.

2. The authors use Deep Interpolation for denoising their calcium imaging data, which improves signal quality but may also introduce temporal artifacts. The supplementary data suggests this processing slightly advances stimulus information and prolongs transient response time. The manuscript should more thoroughly address how this preprocessing might influence the temporal dynamics of the observed responses and potentially affect the avalanche analysis.

Response: We have now added additional analyses that document the specific effects of using Deep Interpolation in our analysis. We now demonstrate no qualitative changes in the response curves for datasets processed without Deep Interpolation (new Supplementary Fig. 8). As previously shown (Capek et al. 2023), our avalanche analysis performed in the data without DeepIP shows a scaling that is maximized with increased coarse-graining ($k = 4$ for the original data, $k = 6$ for the non-denoised version) and achieves a lower scaling exponent (~1.7 vs 2 for the original data).

3. The distinction between genuine network responses and potential direct optical stimulation of non-target cells requires further clarification. While the authors performed a radius exclusion analysis as a control, more detailed characterization of the holographic stimulation profile would strengthen confidence in the specificity of their approach.

Response: We have added new analyses to further demonstrate our precision in stimulating single pyramidal neurons, as well as addressing potential multi-target cases. Please see our response to Reviewer #6 concerns, where this clarification was also requested.

4. The paper lacks a clear discussion of how the observed scaling properties might relate to behaviorally relevant information processing. While the authors demonstrate that network responses encode information about which neuron was stimulated, they do not connect these findings to functional or computational advantages for the animal. This connection would enhance the broader impact of the work.

Response: We have added a new paragraph on Steven's Law that relates the exponents found for the response curves to behavior (Discussion, 5th paragraph). We have also further discussed the consequences of our results for coding/information processing in the brain (Discussion, paragraphs 2-4).

5. Is the statistical approach used to identify positive/negative responders sensitive to the specific threshold? The authors might consider demonstrating robustness across a range of statistical thresholds, or ideally, adopt a more continuous measure of responsiveness that doesn't rely on binary classification.

Response: In our initial submission, we reported the robustness of significant positive/negative responders to a wide range of thresholds (Figs. 2g-j). We have now added statistical correction for multiple comparisons and show that under this contingency, the number of positive responders increases even more for our standard threshold of 2.5% significance, whereas negative responders lose significance under any threshold considered (Fig. 2j). This clearly documents the robust net excitatory response to novel spikes in the cortex.

Reviewer #2 (Remarks to the Author):

This work presents an investigation of the resulting effects on the surrounding network of perturbing the firing of a target neuron in mouse cortex. The authors use optogenetic techniques to stimulate and record the activity of the neurons. The results are substantial and novel, generating interesting takeaways and ideas for future exploration.

Response: We thank the reviewers for their very positive support of our findings.

Most of our feedback is relatively minor and offered to modestly increase the impact of the work with what we believe should be relatively little effort.

Noteworthy results:

1. Injecting spikes into the cortical network causes some neurons to increase their firing rates without decreasing the Fano Factor (as would be expected when driving the network with sensory stimuli). A small population of neurons decrease their firing rates as a result of the injected spikes and some neurons do not significantly change their firing rates.
2. A substantial portion of the neurons in the network respond by increasing their firing rates when the target neuron (TC) spikes, a higher proportion than neurons that are direct postsynaptic targets

of the TC neuron. This is the case whether the target neuron spikes intrinsically (baseline) or spikes in response to optogenetic inputs (response).

3. The network activity following intrinsic spikes (baseline) is significantly different from the response to optogenetically-triggered spikes (response).

A. Populations that do not respond to the optogenetically-triggered TC spikes (NonR) increase spiking in concert with intrinsic TC spikes (this may be caused by both TC neurons and NonR neurons being driven by some third factor, as opposed to the TC neurons driving the NonR neurons).

B. Populations that correlate negatively to the optogenetically-triggered TC spikes correlate positively to intrinsic TC spikes.

C. Neurons that respond positively to optogenetically-triggered TC spikes (PosR) increase their firing rate as the firing rate of the TC neuron increases. However, the increase is actually at a faster rate as a function of intrinsic TC spikes than of novel TC spikes.

In particular, point 3C above to us suggests that there may be an underlying "neural manifold", and that optogenetically-triggered TC spikes are "off-manifold" perturbations, and the circuit limits the extent to which these off-manifold spikes drive activity, similar to other studies.

4. The identity of which neuron was the optogenetically-driven TC neuron can be reliably decoded from the population activity, but this decodability vanishes after about 200 ms.

5. Decodability is more or less robust to changes in the underlying dynamical state of the network, as measured by being in a Low firing or High firing avalanche state.

Novelty and impact

Previously the impact of injecting a single spike into a cortical circuit has been investigated in London et al. (reference 2). The above results significantly extend this in interesting ways that shed substantial conceptual light on the circuit.

Support for results/claims

The presented data does appear to generally support the major claims made in the paper, although we believe additional clarification should be made in terms of the expectations set by previous models, experiments, and basic reasoning. We also make suggestions for additional analysis that we believe would strengthen the support for the major claims.

1. The authors state that the scaling observed in Figs. 2d,e support "efficient" coding. Could you clarify which null model or baseline is used to claim the efficiency? Additionally, please explain how having an exponent that is either too large or too small might be detrimental. Is the 0.5 degree scaling with the intrinsic TC spiking more or less efficient than the 0.3 degree spiking of the induced TC spikes?

Response: We thank the reviewers for this question. We have now replaced our notion of 'efficient' with pointing out the 'robust and widely distributed' nature of how novel spikes are encoded in the network, which is a more accurate description of our findings.

Our new simulation also provides a clarification regarding the response curve exponents (see Fig. 6c), which introduces the reader to the extreme situation of 0 at supercritical dynamics and 1 in the subcritical regime, as outlined in the main text and figure legends. Additional comments on this issue are added to the Discussion (5th paragraph).

2. Related to the preceding point 1, the authors show in Figure 2 critical exponent scaling that they claim should be approximately optimal for propagating information, and in Figure 3 they measure information propagation, but they do not connect the two directly by, say, linking higher or lower critical exponents to more or less decodability.

Response: That is a great suggestion, which we could not explore in our experimental data because, unfortunately, exponents obtained from single targets were too noisy (not enough data) to provide any meaningful relationship with decode-ability. Accordingly, we had exponents by pooling all data. However, in the newly added simulations we now show that the best decode-ability coincides with a system near a critical point (new Figs. 6g, h), which is characterized by response exponents close to those found in the experimental data (new Figs. 6e, f). It might be important to notice that our decoding task is a discrimination task, which might not be very sensitive to how the network responds to different intensities of the target neuron's firing (which is reflective of the system's dynamic range).

3. Related to the preceding point 2, we believe it would strengthen the results to include a version of Fig. 3i for the case of decoding the identity of a TC neuron that emitted a baseline/intrinsic spike. This would help to deduce if the information spreading through the network as a result of the novel spikes spreads more or less efficiently than the information spreading as a result of intrinsic/non-novel spiking.

Response: We thank the reviewers for this suggestion. We have analyzed decode-ability of the same target cells during baseline windows and observed above chance levels, although significantly lower than the ones obtained during the stimulation (modified Fig. 4b, right panel). Unfortunately, it is hard to argue that novel spikes spread more efficiently, since the tasks compared have many differences. For instance, in the baseline task we only used trials in which only one target cell was active, so we could unambiguously label that trial. On the other hand, during baseline activity rate of target cells were many times lower, which would presumably make decoding harder.

4. We believe that the results of the paper would be strengthened by showing the scaling exponents of Fig. 2d,e separately in the Low and High firing regimes of Fig. 4f. Our reason for this is that we feel that Fig 4 attempts to show how the main important results of the preceding figures depend on the global network state, and these scaling exponents appear to be central to the message of the paper alongside the decode-ability.

Response: We thank the reviewers for this suggestion. We have added this analysis in new Supplementary Fig. 11, showing the exponents are robust to the amount of activity in the network preceding stimulation.

5. This point concerns the claims about the network being an efficient propagator of information. A possible alternative interpretation for Fig 2d versus 2e, and Fig. 4i, might be that the network is actually good at quashing the amplitude information of perturbations (as opposed to enhancing these perturbations), as stated on line 126 "network responses to single spikes scale similar to

responses to multiple novel spikes from a single neuron." From this perspective, the injected spikes might be better understood as noise to be suppressed rather than as a signal to be amplified. Again, having more explicit baseline models or expectations in mind may help contextualize the results.

Response: Our decoding analysis confirms that injected spikes are not treated as 'noise' because their corresponding network responses contain information about their origin. We also show that the responses are net excitatory with neurons that demonstrate spike suppression are not significant when multiple comparisons are taken into consideration. We suggest that these findings do not support the interpretation that the network performs a suppression operation on novel spikes.

Soundness and thoroughness of methodology and analysis

While the primary reviewer is not very familiar with the type of methodology used in this study, to the best of their ability it appears to be sound and sufficiently described to be reproducible.

Minor comments

We believe that equations 1 and 2 would be clarified by writing the dependence of the variables on baseline/stimulation, or by writing (baseline/stimulation) on the same lines as the equations. This may want to be carried out throughout the paper, for instance on line 135, where writing that TC and PosR are here the stimulation TC and PosR variables would clarify things significantly.

Response: We thank the reviewers for pointing this out. We have made changes in the text to reflect, when referring to spike counts of a certain population, whether we mean during base or during stim.

When measuring spike scaling, the authors must determine the number of spikes emitted by the target neuron (TC). The authors chose a 132ms window over which to count these spikes. It would be nice to have some justification for this choice of interval.

Response: At a holographic stimulation duration of 100 ms and Dt of 22 ms, we chose an analysis window of five frames, which covers stimulation duration, plus additional time (1 frame) to capture near future stimulus related spiking activity. We have added a reference to the Materials and Methods and Results on the first instance we mention the window and explained it there.

The authors use a threshold alpha to mark populations of neurons which are impacted by the target neuron perturbation, but don't appear to define this in the Methods. Alpha should also be defined first on line 97, and it should be made consistent whether "alpha" or " α " is written in the text.

Response: We thank the reviewers for pointing this out. We have added a reference to Materials and Methods on the first mention and standardized to " α " throughout.

We believe that a more explicit discussion of the advantages of the holographic technique, compared to the results reported in reference 2 (London et al.), would be valuable.

Response: We have added a discussion on the advantages of the holographic technique when compared to alternatives such as microstimulation or intracellular approaches (Discussion, 1st paragraph).

Figure 2d appears to be missing a legend. It may be sufficient to state in the caption for Figure 2d that the legend can be found in 2e.

Response: We have added a note in the caption of panel d saying that it shares a color code with panel e.

The authors frequently state that the power-law scaling is "robust" -- does this refer exclusively to the robustness with respect to alpha as shown in Fig 2g, or is it meant to be taken in a more general sense. If it is more general, what is the justification for this?

Response: It is the first meaning – with respect to alpha. We have clarified it when presenting Fig. 2g results.

What is the definition of $\langle cc \rangle$ just before equation (1)?

Response: We have clarified by adding “average cross-correlation” to the text.

It would aid the reader to clarify that in the caption for Figure 2 the SD is across neurons. This clarification could be done in the Methods.

Response: We have added this to the caption of Fig. 2.

We believe it would be clearer if the lightning bolt in Fig 1c points to the lefthand part of the appropriate interval.

Response: We have changed it accordingly.

Only fewer than 1% of neurons showed negative responses to TC stimulation. Are they genuinely negatively responding or found by chance? It is helpful to clarify how the replicability analysis was conducted in the context of multiple statistical testing.

Response: We thank the reviewers for raising this issue, in line with requests from other reviewers. We added a comment on that on the Materials and Methods subsection on significant responders and also mentioned in the text (Results, 4th subsection, 1st paragraph). Although we have virtually no NegR neurons left after multiple comparisons correction, it is important to note that we are biased towards underestimating them: given our approach to compared spike counts during base vs response and the low (average near zero) base spike counts, it is very hard to find neurons that significantly decrease their spike count during the response. Furthermore, even when loosening the criteria for significance, we see that the population of NegR does indeed anti-correlate with TC firing, maintaining a negative exponent for their response curve.

Reviewer #3 (Remarks to the Author):

Reviewer #4 (Remarks to the Author):

Reviewer #5 (Remarks to the Author):

Ribeiro et al. present a technically impressive study using advanced all-optical stimulation and recording methods to examine how cortical networks respond to novel stimuli. This work continues a relatively recent line of investigation into the coding schemes employed by cortical circuits. As outlined effectively in the discussion, a central challenge in the field is bridging the gap between what is known about the integrative properties of single neurons and the connectivity/activity patterns observed at the network level, and the apparent sensitivity of these networks to small, unexpected inputs.

According to standard synaptic integration models, such small perturbations should exert minimal influence on overall network activity. However, a growing body of experimental work has demonstrated that such perturbations can have amplified, long-lasting effects, even influencing perception and behavior. Understanding this phenomenon holds significant implications for our understanding of neural coding and cortical integration of input.

In this context, both the premise of the manuscript and the methodology used are highly promising. The experiments themselves are technically demanding, and the authors deserve credit for executing them successfully. Moreover, the core finding - that perturbations of as few as 1-7 neurons can be reliably detected at the population level provides compelling evidence that this form of sensitivity is indeed a robust property of cortical networks.

However, the manuscript as a whole is difficult to follow, for several reasons:

1. Conceptual and Terminological Obscurity: The writing lacks clarity and makes liberal use of highly technical terms (e.g., “criticality,” “parabolic neuronal avalanches”) without adequate explanation or motivation.

Response: We have rewritten many parts of the manuscript and added definitions when technical terms were employed. We also added simulations with them aim to shed light into how and why criticality fits into this study.

2. Inadequate Methodological Transparency: There is a lack of raw data presentation, and many key analytic steps are described only superficially. For instance, much of the analysis is done at the spike level, yet no demonstration is provided of how spikes were extracted from the calcium imaging signals.

Response: We thank the reviewer for raising this issue. We have added examples of raw data in new panels (Figs. 1c, d) and in a new figure (Supplementary Fig. 4), where we show calcium traces together with the extracted estimated spikes.

3. Unclear Reference Framework: Some of the results show clear effects, but the lack of a well-defined null model or comparative baseline (e.g., a balanced network with mild external drive) makes it difficult to interpret their significance. At times, I was left unsure whether the paper tackles

deep and subtle questions that are simply hard to understand, or whether it assembles a set of loosely connected findings without a unifying conceptual framework.

Response: To address this issue, and in line with issues raised by Reviewer #1, we have added model simulations to the manuscript, providing a framework on how the experimentally obtained results might relate to balanced (or not) networks under mild drive.

Given these issues, I would recommend that the authors be invited to revise the manuscript, possibly reusing the same data, but with a more transparent, accessible presentation of methods and results. The analysis methods should be explained in greater detail, assuming less prior knowledge from the reader regarding complex neural dynamics.

Specific Comments:

1. Abstract: The abstract is difficult to parse. For instance, “Their influence scales robustly to an exponent between 0.2 and 0.3 relative to their number” is opaque. Similarly, the phrase “This heightened susceptibility... aligns with the behavior of complex systems exhibiting critical dynamics”.

Response: We have modified the abstract accordingly and our added simulations should help the reader understand the more technical parts.

2. Lines 144–148: The discussion of statistical thresholds (α) and PosR identification lacks rigor. Simply lowering the statistical threshold increases the number of detected “responsive” cells, but no control is provided to account for the increased false positive rate. This weakens the argument that the main finding is robust.

Response: We thank the reviewer for raising this point. We have added a false discovery rate correction to the identification of significant responders and show how that relates to the obtained results (see Fig. 2j). We find that at the significance level used, this correction significantly increases the number of positive responders, whereas it removes any negative responders which we now elaborate on in the main text. See also response to Reviewer #6.

3. Lines 156 onward: The Fano Factor analysis is presented without sufficient methodological detail. It’s unclear for which population of cells it was computed (e.g., PosR with default α or those under more permissive criteria). The conclusion — that a single novel spike elicits large-scale responses — seems overstated given the uncertainty in the permissive response criteria. Including examples of raw data (e.g., as in O’Rawe et al.) would greatly enhance credibility.

Response: That analysis (and all other analysis unless specifically mentioned) was done on the 2.5% alpha level – this is mentioned on Fig. 2 caption. We are also now including raw data examples in the supplemental material (Supplementary Fig. 4).

4. Figure 3: The decoding of stimulation targets using population activity is interesting, but the underlying assumptions and statistical controls of the decoding procedure are not clearly

explained. In large recurrent networks operating near criticality, the ability to back-infer the source of stimulation might depend strongly on hidden variables (e.g., direct synaptic links or network topology), which are not discussed.

Response: We have clarified our decoding analysis and also added simulations, which further shed light on how decoding depends on hidden variables (new Figs. 6 & 7).

5. Avalanche Analysis: The treatment of avalanche dynamics and parabolic scaling is not adequately explained in the main text. I found it very difficult to follow both the rationale and implementation of the analysis — even after reading the methods.

Response: We have softened the introduction of our avalanche analysis to better clarify the reasoning behind including them. We also polished their description in the Materials and Methods section.

Minor Comments:

- Figure 1b (top): It is unclear what is being plotted. Is this the z-score of the target neuron? This should be made explicit.

Response: We have expanded Fig.1 caption to state “response spike count z-scored based on baseline count”.

- Line 119: The definition of TC_{sc}^h is unclear. It appears to mean $(\text{TC}_{sc})^h$, but this should be explicitly clarified.

Response: We changed the equation to add the parenthesis for clarification.

- Line 127–130: The discussion in these lines is hard to follow. Rephrasing for clarity is needed.

Response: We have rewritten this part of the results for improved clarity and as a consequence this sentence was removed.

- Line 133: Were these trials excluded from the analysis?

Response: These trials were not excluded from the analysis.

- Line 146: The phrasing regarding α is ambiguous. “Lowering α ” usually means increasing stringency, but here it seems to imply using a more permissive threshold. Please clarify.

Response: We have clarified throughout the text that increased alpha means lowering the threshold for significance.

Reviewer #6 (Remarks to the Author):

Ribeiro et al. aim to characterise the network effects of novel action potential activity in the primary visual cortex, linking this to a potential role in the detection of stimulus novelty. To explore this, they leverage holographically targeted optogenetics, aimed at the stimulation of a single neuron, during spontaneous activity in the visual cortex. They proceed to analyse the effect of stimulation on that neuron, and the changes in activity patterns in the population of simultaneously recorded neurons in the local network. This allows the authors to infer potential functional connectivity within the network and to look at variation in the stimulation driven modulation of network activity in the context of baseline activity levels.

To the best of my knowledge the fact that they are carrying out these manipulations while the animal is not experiencing salient visual stimulus, separates the work from previously published studies reporting similar experiments (Chettih and Harvey, 2019, Russell et al. 2024). It would seem however to somewhat limit the strength of connection they are attempting to supply, between their results and the detection of novelty in real world sensory stimulation, as the brain state will not be comparable.

The methods employed are sound and cutting edge and the results are potentially high impact. I do however have some significant concerns and suggestions for improving the manuscript.

Major

1. The major concern I have is the potential presence of off-target stimulation and the effect this could have on the interpretation of the data. The opsin they are using is mainly soma restricted, however there are still likely to be some off-target effects, and I would like to see this properly characterized. I have not seen this particular method of targeting used before which further adds to the need for resolution quantification.

Response: We understand the reviewer's concern and thank for the opportunity to clarify and strengthen our claims. We have addressed this issue in multiples ways, which are pointed out below.

a. The authors should create physiological resolution plots by varying the location of the stimulation beam relative to the centre of the targeted cells soma (e.g. Marshel et al. 2019). This should be done across X, Y and Z, planes. This should be achievable with the hardware and software they have used. The resolution curve produced should then be used to determine the spatial cut of threshold for excluding any ROIS from their analysis. It is clear from Fig. 2h that the majority of positive followers are in proximity to the target (0~75 microns)

Response: Our experimental setup has been modified since the recording of these datasets, which prevents us from re-establishing the exact stimulation condition to conduct the experiments as outlined in e.g. Marshel et al., 2019. Instead, we have performed a proxy analysis, by taking advantage of the natural movement present in our recordings, with the added advantage that the information is obtained from the actual imaging and stimulation data. We demonstrate that our holographic stimulation response is restricted to a distance below ~5 μm from target center in the x-y plane (new Supplementary Fig. 3).

b. The Z plane will be the most limited resolution in this setup and is likely to stimulate unobserved neurons above and below the target cell. This should be mentioned in the discussion.

Response: We very much agree with this reviewer regarding the extended z-profile in the stimulation laser recruiting potentially more than the target visible in the optical plane. Our experimental design partially addresses this concern: First, we purposely injected low amounts of opsin, resulting in a low density of opsin-expressing cells. This facilitates our ability to stimulate single neurons as the likelihood of recruiting unobserved neurons outside the plane of view is low (see modified Supplementary Fig. 1 for example field of views and opsin-expressing cells density estimation). Second, in our setup the extension of the imaging plane is also increased by not filling the back of the objective completely, again increasing our likelihood to select target cells that are distant from other opsin filled cells within the recording volume.

c. The GG to GR alignment and the Top-Hat profiles shown in supplemental figure 2 are not perfect which furthers the need for this quantification. It would be good to show a larger area on the examples in supplemental figure 3 to better assess off-target stimulation.

Response: We apologize for this confusion. The panel in Supplementary Fig. 2d represents a schematic of the procedure, which is now clearly stated in the legend. In fact, the alignment of the GG to GR is carefully done and checked before experiments to achieve precise stimulation of our target cells, as demonstrated e.g., in our new Supplementary Fig. 3. We agree that the Top-Hat profile is indeed not perfect, as one would expect due to multiple limitations intrinsic to this all-optical setup. For this reason and in the interest of clarity, we provided the original Supplementary Fig. 3 (now Supplementary Fig. 5), which now in addition shows an increased area as requested.

d. In Supplemental figure 3 there are clear examples of stimulation more than one neuron, for example target 27 and 45. There are also large areas of positive fluorescence change around the targets, I am curious how the authors explain this when the net surround has been shown to be inhibitory previously (e.g. Russell et al. 2024). Is it uncorrected neuropil? Off-target activation of dendrites? Artifact from the stimulation laser?

dendrites? Artifact from the stimulation laser?

Response: We thank the reviewer for raising this issue which allowed us to provide a more detailed analysis. Indeed, we have a few clear examples of possible multi-target stimulation. We have analyzed those separately for decoding and show no significant change in the results (see modified Supplementary Fig. 10a).

Regarding the areas around the target, it is expected to occur since we have not corrected for neuropil in this figure. Our aim was to provide as much uncorrected fluorescence information as possible to increase transparency in our stimulation success.

Therefore, we simply calculated pixel-wise stimulus-response DF/F . Dendrite activation should not be a factor given we have soma-targeted opsin. See also our response above demonstrating the spatial precision of our stimulation.

e. The magnitude of the response they report in ‘follower’ neurons looks relatively big compared to previous reports (Fig 1c for example). This increases the concern over any off-target stimulation effects. I am curious why the authors think they see such strong responses resulting from a few spikes in a single neuron. Perhaps it is partially due to the brain state they are studying, this should be discussed more in the manuscript and compared to previous data.

Response: Comparing to previous studies (see a brief list below), our data might have a few key differences that could potentially lead to stronger observed responses, namely a more precise calcium indicator (GCaMP7s) together with the usage of machine-learning-based denoising (Deep Interpolation). Also, the proximity between the TCs and measured neurons compared to some other studies might be an important factor.

Since the reviewer has not pointed to specific papers, we list below example studies on single-cell stimulation and its network effects and conclude a proper comparison with our results is difficult:

- *Chettih & Harvey 2019: hard to compare since they evaluated responses directly from Ca^{2+} traces and perturbed during visual stimulation.*
- *Knauer & Stüttgen 2019: applied juxtacellular stimulation paired with microelectrode array recordings. Smaller effects than us, but cells were much farther away from TCs.*
- *London et al. 2010: 1 spike added = 28 extra spikes in postsynaptic neurons (intracellular stimulus and extracellular recordings, anesthetized rats). Estimated 1 extra spike = 0.04-0.08 Hz local increase in activity. 16 recording sites 50 μm apart (far from TCs).*
- *Meyer et al. 2018: 2PI + patch clamp stimulation. Less than 1% “reliable” followers – more stringent criteria than Positive Responders. Hard to compare since they used OGB as their indicator.*

f. In addition to the resolution quantification, it would be beneficial to display the average movement they are seeing in their preparation. This is relatively easy to do using the values of the motion correction performed in suite2p.

Response: We thank the reviewer for this suggestion. We have added a motion correction analysis in the new Supplementary Fig. 3. There we show that most frames needed no correction at all, with the vast majority needing no more than 1 pixel (0.88 μm) correction.

g. One of the arguments that they are not seeing off-targets near target neurons is that they don’t see a decrease in decoder performance when removing cells within 10 microns of the target. Is this taken from the centroid of the target neuron to the centroid of the non-targeted one? If so how many cells actually fall within this range? It is possible that the lack of decrease in decoder performance is due to a lack of neurons in this area.

Response: Yes, centroid to centroid. We have created a new version of the distance drop plot in which we remove ALL neurons that are located a certain distance from ANY target cell and now also show the number of neurons dropped as function of distance (modified Fig. 4e). There are few neurons dropped in the 0-20 um range, with a small influence in decoding accuracy.

2. In Fig. 1a the authors show, and indeed label, the presence of clear activation of putative dendrites in the imaged FOV. These ROIs are fairly large and can be a large distance from the soma (one is approximately 80 microns away in this example). It is likely that these are part of the target neurons and it is imperative that these ROIs are removed from all analysis. They would produce ROIs with a very high magnitude response and will ensure the possibility of decoding the targeted neuron from the other observed activity. These ROIs need to be removed for the data to be properly interpreted. They could be filtered out using the footprint of the ROI, then potentially the unusually high magnitude of response and extremely high correlation in activity to the target soma. Success should be confirmed manually.

Response: It was our implicit purpose to demonstrate in our original Fig. 1a this issue in these types of all-optical recording experiments (now Fig. 1b). It is because of this potential confounding factor, which this referee is correctly pointing out, that all of our cell selection is manually curated, and the resulting selection does not include dendrites from targets or, in fact, any other cell. We have now clarified this in the Materials and Methods section.

3. Artifact from the stimulation laser is a serious concern with such experiments, particularly when relying on small difference in fluorescence traces for readout. Hopefully there is no issue with the amount of total power being used staying relatively low. It would however be good to confirm this by looking at the time of stimulus onset across all ROIs and looking at the raw fluorescence traces for any sudden increase. If a notable deflection is observed, then the authors may need to attempt artifact subtraction.

Response: We agree that these cross-frequency stimulation artifacts are serious concerns in all-optical experiments. However, no deflections were notable whatsoever with our chosen combination of GECl/opsin, stimulation wavelength (1064 nm) and the low power required to stimulate single targets (<~10 mW). See example raw traces in our new Supplementary Fig. 4.

4. The data is heavily processed which makes it harder to interpret accurately. As a sanity check I would be grateful to see some examples of the same analysis but being conducted using the DF/F rather than the inferred spikes (for example Fig. The authors mention themselves that this process can cause issues, and they have had to remove a small amount of data. Does this impact the number of modulated neurons observed?

Response: We thank the reviewer for this suggestion. We have added a response curve analysis performed directly on DF/F, showing qualitative agreement with our spike results (new Supplementary Fig. 6c). We have also performed the decoding analysis on the calcium traces (new Supplementary Fig. 10e), with predictable lower (given the long decay of calcium traces) but still above chance decoding.

5. The analysis to identify modulated neurons could benefit from correction for false discovery rate, this may reduce the somewhat surprising number of neurons that are being identified as modulated (e.g. Fisek et al. 2023).

Response: As described in Fisek et al. 2023, we have now performed false discovery rate correction to our results. This analysis demonstrates that actually our selected significance level (2.5%) is more strict than the obtained p-values of our datasets would allow us to call significant responders. The FDR-corrected results lead to a number of significant responders that approximately match the one we obtained using a threshold of 10% for significance (~ 70% higher). See Fig. 2j and the Materials and Methods section.

6. I find the plots looking at the transient nature of the decoder performance around stimulation in appropriate for the purpose. Why are they produced in such a way as to give the large increase in performance around the stimulation? (both before and after). How would the decoder possible function before the stimulation of the target cell? In one session they report an increase in performance beginning >200ms prior to the stimulation onset. It would be insightful to see these plots made from rawer data, DF/F rather than inferred spiking, and without the large sliding window they have employed which ruins the temporal resolution.

Response: We thank the reviewer for raising this concern. We have repeated this analysis using a single frame to decode, in order to make the temporal evolution of accuracy more straightforward. As mentioned in the text, the decoder starts working before the stimulation for the same reason the activity ramps up before stimulation: the usage of Deep Interpolation for denoising the data smoothens the time series in time, specially backwards, given the asymmetry in the calcium kernel. For this reason, we had also provided plots comparing the analysis on the same datasets with and without DeepIP. Now with the decoding being calculated on a frame-by-frame basis, it is easy to see that only after the stimulus onset decoding above chance is possible (new Fig. 4f, top). Repeating the analysis with DF/F would not work in this case, given the much lower accuracy (see point above) even when measuring using 6 frames. It also would be a problem for the time after the stimulus is off: given the second-long decay of calcium traces, the influence of the increased activity during stimulation could even bleed into posterior trials.

Minor

It would be good to clarify somewhere that it is more difficult to detect inhibitory effects as these require a significant baseline activity and tend to be lower in magnitude. This relates to the discussion of the brain state, with different levels of activity and inhibition.

Response: We thank the reviewer for pointing this out. We added in the Results (4th subsection, 1st paragraph) and also in the Materials and Methods (subsection on significant responders) comments on the intrinsic problems with detecting decreased firing in bounded, low baseline activity levels.

Figure 2b and others – I can't really distinguish the bars with or without the patterning in a printout of the manuscript. The authors should consider making this easier to distinguish.

Response: We have made the pattern lines thicker, which hopefully will help make them more distinguishable in print version.

Supplemental figure 3 would benefit from a scale bar and the distance being given in μm rather than pixels in the legend.

Response: We have added a scale bar to the figure and added micron scale to the caption.

Line 47 – I am not sure that ‘Optogenetical’ is a word, perhaps optogenetic.

Response: Changed to “holographic perturbation”.

Line 170 – ‘Network’ needs an s on the end

Response: Thanks. Fixed.

REVIEWER COMMENTS

Reviewer #1 (Remarks to the Author):

We have carefully reviewed the revised manuscript and the authors' responses to our concerns. The authors have made substantial improvements to the work, particularly through the addition of rigorous computational simulations that provide important theoretical grounding for their experimental findings. These simulations effectively demonstrate that the observed experimental results align with network dynamics operating near a critical state, significantly strengthening the manuscript's central claims about cortical information processing.

While we acknowledge that some theoretical inconsistencies remain unresolved—specifically the apparent contradiction between obtaining 2D directed percolation scaling exponents from perturbation responses while observing mean-field directed percolation characteristics in avalanche distributions—the authors have provided reasonable explanations for the limitations inherent in two-photon imaging approaches. Their acknowledgment that baseline versus perturbation comparisons are inherently difficult due to correlated ongoing activity is appropriate, and their argument that the power-law response curves themselves provide evidence of criticality is acceptable within the broader context of their findings.

The experimental work remains technically sophisticated and methodologically sound. The core findings regarding single-neuron perturbation effects and their propagation through cortical networks represent novel and significant contributions to our understanding of cortical coding mechanisms. The demonstration that individual action potentials can reliably encode information about their origin across distributed networks, despite high variability and ongoing avalanche dynamics, provides important insights into cortical information processing. The authors have adequately addressed our concerns about experimental controls, methodological transparency, and the need for theoretical framework. The addition of simulations particularly strengthens the work by providing mechanistic explanations for the observed phenomena and demonstrating the relationship between network criticality and information transmission efficiency.

We recommend acceptance of this manuscript. The work makes meaningful contributions to neuroscience despite remaining theoretical complexities, and the experimental and computational findings will be of significant interest to the broader community studying cortical dynamics and neural coding.

Response: We appreciate the reviewers' support of our experimental findings and encouragement to strengthen our criticality claims through simulations. Their suggestions and insightful comments substantially improved our work.

Reviewer #2 (Remarks to the Author):

1. We thank the reviewers for their thoughtful and substantial response and revision. We believe that the addition of the accuracy plot in Figure 4b is very interesting and really contributes to the investigation of “novel scaling of criticality”, since we feel that an investigation of novel spikes should compare against non-novel spikes. The addition of Figure 6 also helps to

provide a connection between the 0.285 scaling observed in experiments and purported criticality, and to establish the functional significance of this criticality (encoding novel inputs in a way that can be more successfully decoded).

Response: We thank the reviewers for their very positive support of our findings and focus on differentiating decoding between novel vs. non-novel, i.e. spontaneous, spikes.

2. Regarding the new Figure 6, we do have some reservations about how universal this correspondence between the 0.285 exponent scaling and criticality is among appropriate networks for modeling cortex, especially since even very small deviations appear to correspond with networks outside of the critical regime (inset of Fig. 6f). In addition, Supplementary Figure 11c indicates that the scaling relationship breaks down in the higher firing rate regime, unlike what is experimentally observed in 11a. If the network simulations require fine-tuning, we feel that this should be pointed out, and in general the robustness to changes in model parameters of the 0.285 exponent scaling at criticality should be discussed.

Response: We agree with the reviewers that our simple model follows key ideas in critical dynamics. However, it does not fully explain the robust scaling seen in our experimental data, such as scaling that remains consistent regardless of overall network activity, and the fact that the scaling value in the model is highly sensitive to fine-tuning near the critical point. We have now added these concerns to the discussion and suggest that including inhibition in cortical models may help create more robust models that better reflect our experimental findings.

In addition, we believe that there is still one more piece remaining to fully connect novelty, scaling, criticality, and functional significance (novel stimulus decodeability). As in the new accuracy plot for Figure 4b, we believe that the key comparison between the base and response cases should be extended to Figure 6, in order to show the effect of novel spikes in the simulated model. We believe this would entail the following: a version of Fig. 6c where the abscissa is the firing rate of non-TC neurons; a version of Fig. 6e where the target count is replaced by the number of non-TC spikes; a version of Fig. 6f where the slope γ is measured for the non-TC spikes; and most importantly, versions of 6g and 6h where the accuracy and F1-scores are measured when decoding non-TC spikes. Taken together, this should show that novel events (as opposed to non-novel events) are best encoded by critical networks, and that appropriate network models of the cortex generically exhibit ~ 0.285 exponent scaling at criticality.

Response: We now extended our key comparison between the base and response cases to Figure 6. Specifically, we: 1) Created versions of Fig. 6e and 6f for the baseline (although still from TCs, but it was important to evaluate from the same neurons and their connectivity structure to the rest of the network), SFig. 13, which shows qualitatively similar results to the experimental data (presence of power law scaling responses even during baseline, with modified exponents –related to the amount of correlations in the network); and 2) Replicated our experimental result in subpanel Fig 4b (right), now Fig. 4c, in our simulations. We show in Fig. 6g (see also new SFig. 14a) that there is a significant increase in decoding of novel spikes at criticality compared to spontaneous, non-novel spikes (in fact non-novel spikes lead to chance level decoding).

We think it would also be beneficial to extend Figures 6f, g, and h to sigma values in the supercritical regime, and to add two more sigma values in Figure 6e to show (nearly) flat lines more fully in the supercritical and subcritical regimes (perhaps sigma values of 0.9 and 1.01 to correspond to those chosen elsewhere).

Response: We have now added results for sigma for the supercritical regime and demonstrate that decodability is maximal at criticality. We would like to point out that decode-ability naturally gets harder as there are more unrelated spikes in the network. This is why it was important for us to set the external drive of each network so that the match the activity level of the experimental data during baseline. However, note that the supercritical regime is marked by a transition to a state where activity self-sustains: Even in the absence of drive, activity continues on a characteristic level, above what we needed to match experimental levels (see Fig. 6c, orange curves). Thus, the only way we could match activity levels in the supercritical regime was to rely on: 1) extremely low Poisson drive and 2) finite size effects. Together, those combine into the raster displayed in Fig. 6b (bottom): activity naturally dies out (due to the finite size of the network and low rate of drive, there is a non-zero chance avalanches will end “prematurely”) and it takes a long time for it to restart (due to very low driving rate). These conditions, on the other hand, lead to supercritical decoding being trivial in a large fraction of trials, in which the network was completely quiet before stimulation. We have now introduced decoding by removing trials with those “artificial” low levels of activity (Fig. 6g and SFig. 14b, c).

Regarding adding more sigma values to Fig. 6e, we have not done that originally because it would make the figure too busy, and the summary result (obtained exponents) is already shown for all branching parameter level in Fig. 6f. We believe adding more curves in that panel would be detrimental to the result we want to highlight.

Please add to the model description the mechanism that shuts down the bursting activity in the supercritical network after approximately eight seconds in Fig. 6b.

Response: The above arguments that explain the pauses observed for the supercritical simulations were added in the relevant subsections of the text.

It seems to us that the question of mechanistically why novel spikes are more decodable than non-novel spikes would be an interesting investigation. The authors may want to add such a question and insights from the model to the Discussion.

Response: We have added a discussion on decoding of novel spikes versus baseline spikes.

In Fig 6c and 6e, the dashed lines should be clearly defined in the figure caption.

Response: As to not confuse these slope values with sigma values in the same plot, we in addition describe those slope values now clearly in the figure legend.

Reviewer #3 (Remarks to the Author):

I co-reviewed this manuscript with one of the reviewers who provided the listed reports. This is part

of the Nature Communications initiative to facilitate training in peer review and to provide appropriate recognition for Early Career Researchers who co-review manuscripts.

Reviewer #4 (Remarks to the Author):

Reviewer #5 (Remarks to the Author):

First, thank you for the very substantial revision. Responding coherently to six reviewers is no small feat; the manuscript is much clearer and tighter as a result. Almost all of my original points were addressed satisfactorily.

Response: We thank this reviewer for her/his generous acknowledgement of our efforts and work.

However, I would like to highlight one point that may not have been explicitly enough addressed in my previous report and that still remains unaddressed. In my last report, I noted that the absence of reference and null models makes it difficult to evaluate some of the results. Specifically, I want to focus on the avalanche analysis. While the exposition and parabolic shape results are clearer now (Fig. 5b–d, f–h), the claim would be stronger if you included a surrogate/permutation analysis. This would help demonstrate that scaling is not an artifact of thresholding, finite windows, or non-stationarity. I suggest running the same analysis pipeline on null models that preserve low-order statistics but break the cascade structure. For example, you could try one of the following: Circular time-shift of the stimulation train, or a Trial/epoch block-shuffle of it, or an event-time jitter ($\pm 1, 2$ bins). I'd expect that the power-law size/duration would weaken or fail under these nulls, which would strengthen your findings.

Response: We have now added circular-shift shuffle results in a new Supplemental Material Figure 11. These results demonstrate a significant steeper (non-power law) size and duration distribution compared to the original avalanche dynamics and importantly, an abolishment of parabolic avalanche organization as evident in the slope $\gamma \ll 2$ and staying close to 1.3 for all temporal coarse graining (SFig. 11c, d). These findings are in line with previous randomization/permutation/shuffle results on parabolic avalanches identified in 2-photon imaging from awake animals (Capek, Ribeiro et al., Nat Comms, 2023).

Additionally, it would be helpful to include a one-line limitation statement in the discussion. It should note that because the avalanche analyses are computed within finite imaging epochs and fields of view, the duration and size distributions are truncated and can be biased by these finite-window effects. This would help remind readers that the exponents should therefore be interpreted with caution.

Response: We fully agree with the reviewer that the methodological constraints in 2-photon imaging limit the range of our distributions and as such measures of slope values to estimate exponents. We have now added the following sentence in the Discussion: "Because our avalanche analyses were computed within finite imaging epochs and fields of view, the duration and size distributions are truncated and can be biased by these finite-window effects. Therefore, their corresponding estimated exponents should be interpreted with caution."

Reviewer #6 (Remarks to the Author):

Thank you for addressing the comments and for all the hard work.

Response: We thank this reviewer for the detailed and constructive comments allowing us to further clarify essential aspects of our experimental paradigms and potential pitfalls in interpretation.

REVIEWER COMMENTS

NCOMMS 25-08029B

Reviewer #2 (Remarks to the Author):

We thank the authors for their revision. We believe that our concerns were adequately addressed and just have a few minor points for improving clarity.

In Figure 4b, please specify which bar corresponds to accuracy and which to F1 score.

We have added this information in the legend.

In Figure S13a, please write in or state in the caption the slope value for the dashed line.

We have added a statement in the legend.

In Figure S14b, please define what threshold is more explicitly.

We have added a sentence to clarify what the threshold is in the legend.

In Figures S6g and 14b, please specify that accuracy is with respect to decoding the TC spike.

We have specified in those two instances that we are referring to TC decoding on the respective legends.

We agree that it would be interesting for future work to explore using inhibition, such as a negative Poisson drive, in the supercritical regime. As is we are satisfied with the approach taken here for the scope of this work.

Reviewer #3 (Remarks to the Author):

Reviewer #5 (Remarks to the Author):

The authors have answered all my concerns appropriately. I recommend publication.

Congratulations on a remarkable work that combines state-of-the-art experimental and theoretical methods. I still find it challenging to wrap my head around the decoding findings. I hope it will steer enough attention to encourage further studies into it.